# Aberrant survival of hippocampal Cajal-Retzius cells leads to memory deficits, gamma rhythmopathies and susceptibility to seizures in adult mice

Martina Riva[1,2], Stéphanie Moriceau[3], Annunziato Morabito[4,12], Elena Dossi [5,12], Candela Sanchez-Bellot[6,12], Patrick Azzam [1,2,12], Andrea Navas-Olive [6], Beatriz Gal[6,7], Francesco Dori[1,2], Elena Cid [6], Fanny Ledonne[2], Sabrina David[8], Fabrice Trovero[8], Magali Bartolomucci [9], Eva Coppola[2], Nelson Rebola[4], Antoine Depaulis[9], Nathalie Rouach [5], Liset Menendez de la Prida [6], Franck Oury [10] & Alessandra Pierani [1,2,11] ✉

Cajal-Retzius cells (CRs) are transient neurons, disappearing almost completely in the postnatal neocortex by programmed cell death (PCD), with a percentage surviving up to adulthood in the hippocampus. Here, we evaluate CR's role in the establishment of adult neuronal and cognitive function using a mouse model preventing Bax-dependent PCD. CRs abnormal survival resulted in impairment of hippocampus-dependent memory, associated in vivo with attenuated theta oscillations and enhanced gamma activity in the dorsal CA1. At the cellular level, we observed transient changes in the number of NPY+ cells and altered CA1 pyramidal cell spine density. At the synaptic level, these changes translated into enhanced inhibitory currents in hippocampal pyramidal cells. Finally, adult mutants displayed an increased susceptibility to lethal tonic-clonic seizures in a kainate model of epilepsy. Our data reveal that aberrant survival of a small proportion of postnatal hippocampal CRs results in cognitive deficits and epilepsy-prone phenotypes in adulthood.

Cognitive impairments and epilepsy are common comorbidities in the majority of patients with abnormal brain development, including cortical malformations and autism spectrum disorders[1]. While cellular processes like neurogenesis, migration, synaptogenesis or myelination are recognized building blocks of circuit formation, an emerging player in the assembly of cortical networks is programmed cell death (PCD). In the nervous system, PCD finely tunes the density of neuronal populations and their targets. About 20-30% of the prenatally over-produced neurons die as they fail to compete for survival signals and to integrate into neuronal networks shortly after birth[2,3]. In addition,

[1]Université Paris Cité, Imagine Institute, Team Genetics and Development of the Cerebral Cortex, 75015 Paris, France. [2]Université Paris Cité, Institute of Psychiatry and Neuroscience of Paris, INSERM U1266, 75014 Paris, France. [3]Platform for Neurobehavioral and metabolism, Structure Fédérative de Recherche Necker, 26 INSERM US24/CNRS UAR, 3633 Paris, France. [4]Sorbonne Université, Institut Du Cerveau-Paris Brain Institute-ICM, Inserm U1127, CNRS UMR 7225, 47 Boulevard de l'Hopital, 75013 Paris, France. [5]Center for Interdisciplinary Research in Biology (CIRB), College de France, CNRS, INSERM, Labex Memolife, Université PSL, Paris, France. [6]Instituto Cajal, CSIC, Madrid, Spain. [7]Universidad Camilo José Cela, Madrid, Spain. [8]Key-Obs SAS, 13 avenue Buffon, 45100 Orléans, France. [9]Université Grenoble Alpes, Inserm, U1216, Grenoble Institut Neurosciences, 38000 Grenoble, France. [10]Université Paris Cité, CNRS, INSERM, Institut Necker Enfants Malades-INEM, 75015 Paris, France. [11]GHU Paris Psychiatrie et Neurosciences, Hôpital Sainte Anne, 75014 Paris, France. [12]These authors contributed equally: Annunziato Morabito, Elena Dossi, Candela Sanchez-Bellot, Patrick Azzam. ✉e-mail: alessandra.pierani@inserm.fr

proper cortical development also depends on the action of specific cell types that stay transiently during the construction of neural circuits. These transient cell types include Cajal-Retzius cells (CRs), subplate neurons, cortical plate transient neurons and the first wave of embryonic oligodendrocyte precursors. These cell populations are unique, as they almost completely disappear in the neocortex towards the end of cortical maturation shortly after birth in mice[3–13].

Although still controversial, the persistence of CRs during advanced postnatal life is reported in different pathologies[14]. In particular, abnormal persistence of transient cells with molecular signatures of CRs is found in cases of temporal lobe epilepsy (TLE), Ammon's horn sclerosis, focal cortical dysplasia and polymicrogyria[15–17]. The anomalous distribution of CR subtypes in cortical layer 1 and the presence of neurons in the white matter of the prefrontal cortex in schizophrenic patients have also been attributed to altered PCD of transient cells[18,19]. Together, this evidence raises the intriguing hypothesis that the maintenance/persistence of transient cell populations contributes to cortical circuit dysfunction.

CRs are amongst the first-born neurons in the developing cerebral cortex and comprise three molecularly and functionally distinct subtypes identified by their spatial origins: septum, hem and pallial–subpallial boundary (PSB)[20]. At embryonic stages, they play roles in controlling migration of excitatory and inhibitory neurons, as well as the phenotype of precursor cells, the development of hippocampal connections and the size of cortical areas[21–26]. Notably, before their number is reduced by the end of the second postnatal week in the neocortex, CRs are embedded into immature circuits where they mainly receive GABAergic synaptic inputs, suggesting a role in early cortical connectivity (for reviews, see refs. [12,14,22]). Strikingly, developmental decline of hippocampal CRs is slower than that in the neocortex, with roughly 30% of CRs remaining integrated into the adult microcircuit[27,28].

Recently, we developed a murine model (i.e. *BaxCKO^ΔNp73Cre*, herein *BaxCKO*) in which CRs death is prevented by invalidation of the pro-apoptotic factor Bax in a Cre-specific manner targeting both septum- and hem-derived CRs. Using this model, we found that only septum-derived neocortical CRs can be rescued from PCD in a Bax-dependent manner[13]. In the somatosensory cortex of *BaxCKO* juvenile mutants, layer 2–3 pyramidal neurons displayed hypertrophy and increased number of dendrites and spines, resulting in an enhanced excitatory/inhibitory (E/I) ratio, and hyperexcitability of upper layer circuits[29]. Nevertheless, whether hippocampal CRs undergo Bax-mediated PCD and whether their persistence throughout development and into adulthood may lead to behavioral alterations and pro-epileptic phenotypes remained unexplored. To address these questions, we exploit the *BaxCKO* model to evaluate the impact of preventing CR demise in adults.

## Results

### Altered hippocampal-dependent behavior in *BaxCKO* mutant mice

In order to test whether the persistence of transient CRs could cause changes in cognitive functions along the animals' lifespan, we first performed behavioral tests on *BaxCKO* mice and control littermates at different ages. All throughout the paper we will examine juvenile (3–5 postnatal weeks), young adult (6–9 postnatal weeks) and adult mice (>3 months old). A primary screen of 85 tests[30] designed for a global assessment of physical status and physiological parameters (like motility, senses, spontaneous behaviors, reactivity, reflex, fear) was performed on both male and female juvenile mice. We found no difference between juvenile *BaxCKO* mutant mice and control littermates (Table S1). When we tested young male adult *BaxCKO* mice in the Cookie test, they displayed normal latency in finding and eating the hidden biscuit in both a first and second assay at a 2 min-interval, indicating no defects in olfaction and motivation (Fig. 1a upper panels). When tested 24 h later, only control mice exhibited a shorter latency in

finding the biscuit, suggesting mild impairment of short-term memory in *BaxCKO* mutants (Fig. 1a lower panels). Nevertheless, when young adult mice were tested in the Y-maze, no significant difference between genotypes was detected in the number of entries in the arms and percent of alternation (Fig. 1b), indicating that their immediate working memory was intact. Using various tests, we also assessed exploratory behavior, motor coordination, sociability and motivation in young *BaxCKO* adult males. We did not find any abnormalities in the open-field, elevated plus maze, 3-chambers, forced swim test or Rotarod when compared to control (Fig. S1a–e). Altogether, these results indicate that juvenile and young adult *BaxCKO* mice display normal balance and motor coordination, immediate working memory and/or willingness to explore. However, mutants exhibit spatial memory impairments at young adult age.

To understand whether this mild memory impairment observed in young adult *BaxCKO* mice could persist or even worsen along life, we evaluated hippocampal-dependent memory function in adult mice. Using a modified version of the Novel Object Recognition (NOR) paradigm[31,32], we first measured the rodents' ability to recognize a novel object in the same environment. As expected, we found that control male mice exhibited preference for novel versus familiar objects (Fig. 1c). In contrast, male *BaxCKO* mutants were unable to discriminate the novel object (Fig. 1c), suggesting that persistence of CRs could alter episodic memory in adults. Performing the Morris Water Maze (MWM) test[33,34], we next found that male *BaxCKO* mutants required a longer time to find the hidden platform when compared to their control littermates (Fig. 1e), indicating alterations of hippocampal-dependent spatial memory. We next confirmed this observation, using the novel-object location (NOL) test[35–37], an efficient behavioral task assessing hippocampal-dependent spatial learning and memory. We found that *BaxCKO* mutant mice explored the relocated object significantly less than control littermates during the testing phase (Fig. 1d). Similar results were obtained for adult females (Fig. S2a–c). However, when assessing fear memory using contextual fear conditioning (CFC)[38], we did not observe any changes in baseline freezing time or in context-elicited freezing time during the testing phase in both male and female adult *BaxCKO* mutants as compared with their control littermates (Figs. 1f and S2d). The retrieval of contextual fear memory is controlled by a hippocampal–amygdala pathway. Our observation indicates that *BaxCKO* mutants can correctly associate the context to the aversive stimulus and to subsequently extinguish fear, similar to control littermates. By contrast to the spatial memory, these results suggest that the retrieval of contextual fear memory, controlled by a hippocampal–amygdala pathway, is not altered in our mutant mouse model. Finally, mutant male and female adult mice did not exhibit any alteration in anxiety- and exploratory-like behaviors (Figs. S1f, g and S2e,f).

Altogether, our behavioral data indicate that while persistence of transient CRs results in only a mild learning and memory phenotype in young adult animals, it severely impacts learning and memory in adult mice. These results suggest that the aberrant persistence of otherwise mostly transient CRs may induce a major impairment of hippocampal-dependent activity and cognitive functions.

### CRs survive in the juvenile and adult hippocampus

Previous reports have shown that CRs populating the hippocampus derive from the cortical hem[28,39]. We had previously found that *BaxCKO* mutants exhibit an aberrant survival of septum-, but not hem-derived CRs, in the medial and lateral cerebral cortex[13] and, in particular, in the juvenile S1 neocortex[29]. Surprisingly, behavioral defects (Fig. 1) suggested an important role of hippocampal circuits and led us to entertain the possibility of a persistence of hippocampal CRs in *BaxCKO* mutants. Thus, we analyzed whether hem-derived CRs survive in the hippocampus, unlike their ontogenically related CRs in the neocortex. We first quantified CRs in *BaxCKO* mutants using

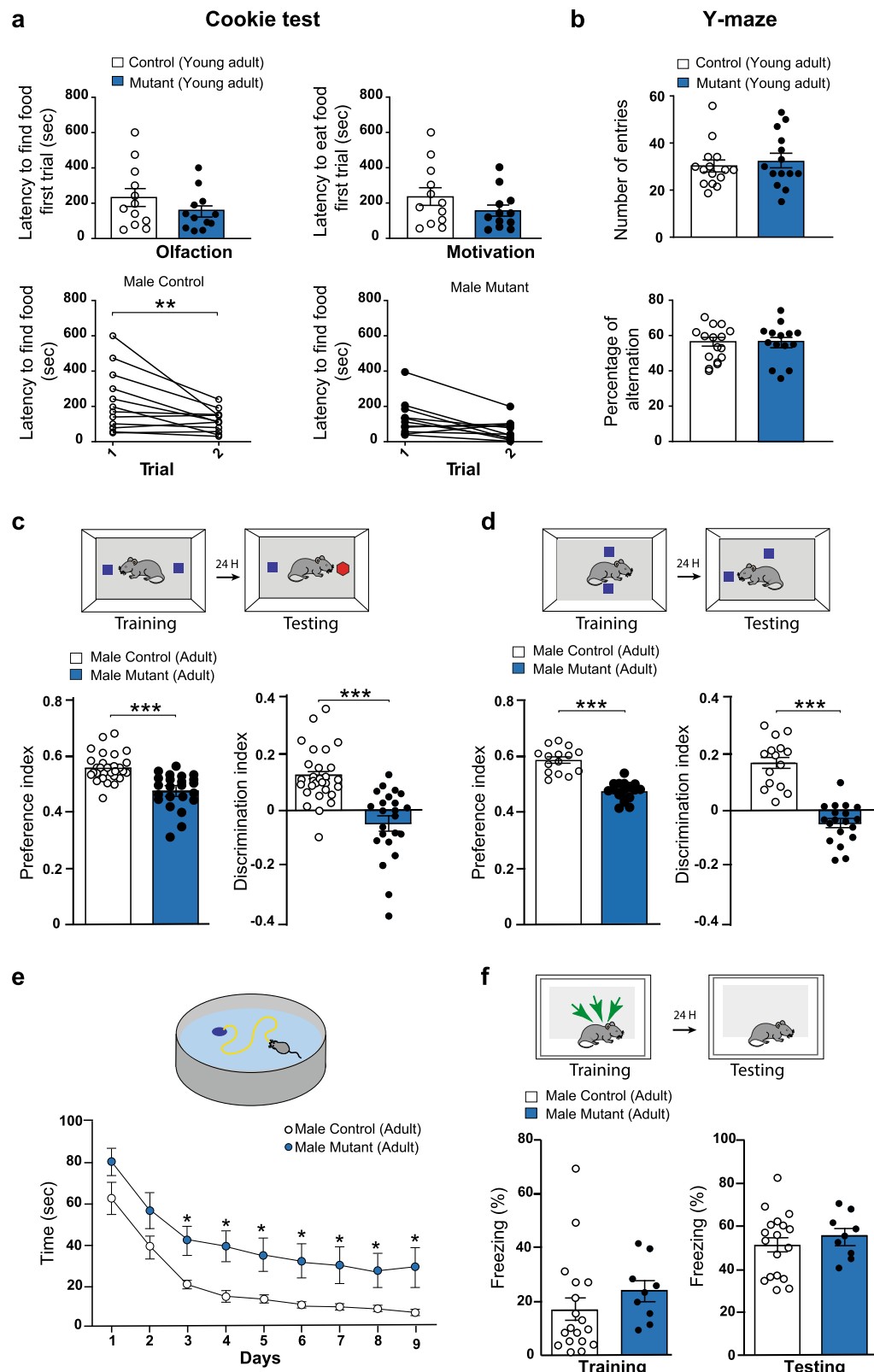

the *ΔNp73Cre*-dependent expression of the *Tau*^*GFPiresLacZ* reporter along the hippocampal fissure (HF), including CA1/CA2/CA3/dDG (dorsal or suprapyramidal blade of the DG), and vDG (ventral or infrapyramidal DG). Our quantification of CRs along the HF did not allow for discrimination of their positioning between the *stratum lacunosum moleculare* (SLM) of the CA1 and the outer molecular layer (OML) of the DG. This led us to group the dDG into the HF and the vDG was

treated as a separate area. Unexpectedly, we found a 27% increase in the density of CRs in juvenile (Postnatal day 24 (P24)) *BaxCKO* mice compared to controls in the HF, but not in the vDG (Fig. 2a). Persistence of CRs (30%) was still observed in the HF of *BaxCKO* adults and absent in the vDG (Fig. 2b, c). Increased numbers of CRs were also detected in adult mutant animals by immunostaining for P73, a known CRs marker[40] (Fig. 2d bottom right panel). To confirm that the

**Fig. 1 | BaxCKO mutant male mice exhibit deficits in hippocampal associated memory. a** Cookie test on young adult (9 weeks old) male animals. No differences in the latency to find or eat the hidden food in the first trial (Trial 1) between controls ($n = 11$) and mutant ($n = 11$) animals (upper panels) indicating normal olfaction and motivation ($p = 0.2913$ and $p = 0.2657$, respectively, Mann–Whitney test). When the test was repeated 24 h later (Trial 2), mutants did not display a shorter latency to find the food ($p = 0.1141$, lower panels) contrary to controls ($p = 0.0044$) (RM-two way ANOVA, Sidak's multiple comparison test), indicating an impairment in spatial memory (mutanst $p = 0.0539$ and controls $p = 0.0271$ according to Wilcoxon matched-pairs signed rank test). **b** Y-maze test on young adult male animals showing no differences in the number of entries in the arms ($p = 0.6912$) or percentage of alternation ($p = 0.9910$) between controls ($n = 14$) and mutants ($n = 14$, Mann–Whitney test). **c.** Novel object recognition (NOR) test on adult male mutants ($n = 23$) and control littermates ($n = 28$). Preference and Discrimination index were measured 24 h after the training phase to assess memory performance. Mutant animals showed impairments in discriminating the new object (Preference Index: controls $0.562 \pm 0.009$; mutants $0.478 \pm 0.013$,

$p < 0.0001$; Discrimination Index: controls $0.124 \pm 0.019$; mutants $-0.044 \pm 0.026$, $p < 0.0001$, Mann–Whitney test). **d** Novel object location (NOL) test on adult male mutants ($n = 19$ males) and their controls littermates ($n = 15$). Preference and Discrimination index were measured 24 h after the training phase. Mutant animals showed impairments in discriminating the new located object (Preference Index: controls $0.586 \pm 0.01$, mutants $0.474 \pm 0.008$, $p < 0.0001$; Discrimination Index: controls $0.172 \pm 0.023$, mutants $-0.050 \pm 0.017$, $p < 0.0001$; Mann–Whitney test). **e** Morris Water Maze (MWM) test on adult male ($n = 19$ mutants and $n = 15$ controls) mice for 9 days. Mutant animals required more time to find the hidden platform ($F(1, 31) = 6.198$, $p = 0.0184$ for Genotype, RM Two-way ANOVA). **f** Contextual Fear Conditioning (CFC) in adult males ($n = 9$ mutants and $n = 18$ controls). Percent freezing time was recorded during the training (as control for basal level, $p = 0.0757$, Mann–Whitney test) and testing phases (to assess memory performance, $p = 0.5612$, Mann–Whitney test). No differences were detected between control and mutant animals. Asterisks indicate statistical significance (*$p < 0.05$; **$p < 0.01$; ***$p < 0.001$). All data are presented as mean ± SEM.

genetically traced surviving cells were indeed CRs, we used a $R26^{Tom}$ reporter that showed a specific increase of TOM⁺ cells in mutant animals along the HF, but not in the vDG (Fig. 2e). Immunostaining experiments confirmed that all the TOM⁺ cells expressed RELN, but not GABA (Fig. 2e). Lastly, in order to determine whether surviving CRs were hem-derived, we quantified CRs in $BaxCKO^{Wnt3aCre}$ ($BaxCKO^{Wnt3a}$) animals using Cre-dependent expression of the $Tau^{GFPiresLacZ}$ reporter tracing cells derived from the cortical hem. We detected an enhanced number of βgal⁺ CRs along the HF in juvenile $BaxCKO^{Wnt3a}$ mutants with respect to controls indicating that the persistent CRs observed along the HF derived from the hem (Fig. S3a). Although all persistent CRs expressed RELN, no altered RELN levels were detected in the hippocampus of adult mutant animals (Fig. S3b). We conclude that PCD of a portion of hem-derived CRs in the hippocampus is Bax-dependent, in contrast to the neocortex where no difference in CR persistence was observed in $BaxCKO^{Wnt3a}$ mice[13]. Altogether, these experiments indicate that hem-derived CRs in the hippocampus persist in $BaxCKO$ mutants, leading to altered hippocampal-dependent behaviors.

### BaxCKO mutants exhibit hippocampal rhythmopathies

We next evaluated whether the persistence of CRs caused functional changes in the dorsal hippocampus, given the major role of this region in spatial and episodic-like memory. To this purpose, we recorded male and female $BaxCKO$ mutants and their control littermates using the awake head-fixed preparation[41]. This approach permitted examination of laminar local field potential (LFP) changes during periods of running and immobility using 16-channel linear silicon probes (Fig. 3a). LFP signals were obtained over several days from the different hippocampal layers, and the probe tracks were verified to confirm comparable proximo-distal locations (Fig. 3b; 17 sessions from 3 control, 16 sessions from 3 mutant animals).

During theta activity accompanying running, we found strong differences between control and mutant electrophysiological profiles in terms of the amplitude of oscillations at both the theta (4–12 Hz) and gamma bands (40–80 Hz). The theta band power was significantly attenuated at the SLM of $BaxCKO$ mice, consistent with the expected location of persistent CRs (Fig. 3c; see quantification at bottom). Instead, the power of gamma oscillations was significantly increased both at the SLM (Fig. 3c) and the *stratum pyramidale* (SP) (Fig. 3d, bottom). Neither of these changes were explained by differences of running speed between groups ($p = 0.264$, Kruskal–Wallis test). In hippocampal sections containing the probe track of individual penetrations (experimental sessions) we could examine the histological correlates of oscillatory changes. Using immunostaining against P73, we confirmed the presence of CRs along the HF, which was elevated in the mutant group (Fig. 3e, bottom). The density of CRs along the HF across animals significantly correlated with the increase in gamma

power at the SP (Fig. 3f, top) and the attenuation of theta oscillatory power at the SLM (Fig. 3f, bottom), suggesting that an increase in hippocampal CR density may determine the emergence of theta/gamma rhythmopathies in adulthood. The laminar specificity of changes caused by persistent CRs can be related to different spatially segregated microcircuit determinants of theta and gamma oscillations[42]. No changes were detected in other relevant hippocampal oscillations accompanying immobility periods, such as sharp-wave ripples (power: $p = 0.0728$; frequency: $p = 0.749$).

Therefore, spatial and episodic-like memory deficits in $BaxCKO$ mutants are associated with underlying alterations of the theta and gamma activities, supporting rhythmopathies deleterious for memory function, as reported before in rodent models of TLE[43,44].

### NPY⁺ interneuron density is enhanced in CA1 upon CRs persistence

Several developmental processes are concomitant with hippocampal CRs demise and our LFP data suggest that gamma rhythmopathies in $BaxCKO$ mutants emerge from microcircuit defects. Given the role of GABAergic interneurons in gamma oscillations, we thus probed whether interneuron subtypes were altered in the hippocampus of $BaxCKO$ mutants upon CR persistence. We first evaluated the total number of interneurons and their distribution using in situ hybridization (ISH) for GAD67, a rate-limiting enzyme involved in GABA synthesis. We detected no differences in the density of GAD67⁺ interneurons across all hippocampal regions of adult $BaxCKO$ mice compared to controls (Fig. 4a). However, when we repeated the experiment in juvenile animals, we observed an increased density of GAD67⁺ cells in the CA1/CA2, but not in the CA3 or DG regions, of mutant mice compared to controls (Fig. 4b), with no difference in the total number of hippocampal interneurons ($p = 0.3429$). This suggests that abnormal survival of CRs promotes transient changes in GABAergic neuron density in the CA1/CA2 region of juvenile $BaxCKO$ mice, which is no longer apparent in adulthood.

In the hippocampus, CRs in the SLM receive local GABAergic inputs from neuroglial form (NPY⁺) as well as from *oriens lacunosum moleculare* (OLM) (SST⁺) interneurons[45]. We thus analyzed the number of SST⁺ and NPY⁺ interneurons present at juvenile and adult ages. We found no difference for SST⁺ interneuron density in CA1/CA2, CA3 and DG or in their total number in both adults and juveniles (Figs. 4c and S4a). In contrast, a specific enhanced number of NPY⁺ cells in CA1/CA2 was observed in juvenile $BaxCKO$ mutants, but not in DG or CA3, resulting in an increase in the total number of NPY⁺ interneurons (Fig. 4d). However, no differences were detected in adult $BaxCKO$ mice (Fig. S4b).

Together, these results show that the enhanced number of hem-derived hippocampal CRs that survive up to adulthood is accompanied by a transient outnumbering of GABAergic interneurons, particularly

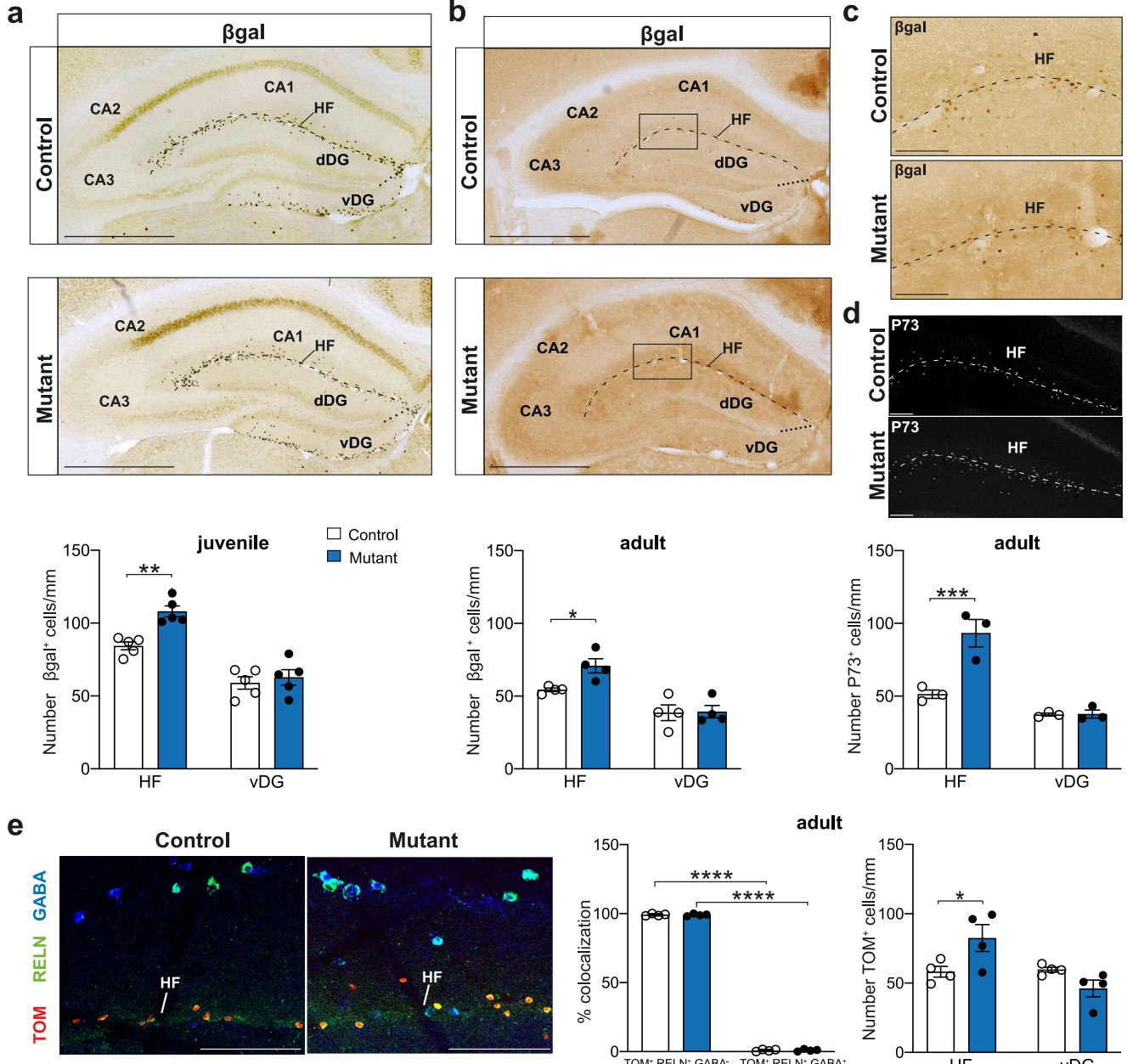

**Fig. 2 | Enhanced survival of CRs in the juvenile and adult hippocampus.**
**a** Coronal sections of juvenile (P24) control and *BaxCKO* (samples carry *Tau^nlsLacZ* reporter) hippocampus immunostained for βgal (top panels). Quantification of βgal⁺ cells in the different hippocampal regions of control and mutant animals (bottom panel) showing increased number of CRs in the mutant hippocampal fissure (HF) (84.34 ± 6.06 cells/mm in controls versus 108.06 ± 8.42 cells/mm in mutants; *p* = 0.0018) but not vDG (58.94 ± 9.5 cells/mm in control versus 62.75 ± 11.85 in mutants; *p* = 0.7709) (Two-way ANOVA, Sidak's comparison test; *n* = 5 controls and *n* = 5 mutants). Scale bar: 500 μm. **b** Coronal sections of adult control and *BaxCKO* (samples carry *Tau^nlsLacZ* reporter) hippocampus immunostained for βgal (top panels). Quantification of βgal⁺ cells in the different hippocampal regions of control and mutant animals (bottom panel) showing increased number of CRs in mutant HF (54.32 ± 2.41 cells/mm in controls versus 70.85 ± 9.75 cells/mm in mutants) but not in vDG (38.61 ± 10.73 cells/mm in controls versus 39.29 ± 8.52 cells/mm in mutants, *p* = 0.0348 and 0.9922, respectively, two-way ANOVA, Sidak's multiple comparison test; *n* = 4 controls and *n* = 4 mutants).

The dashed line represents the HF including SLM of CA1/CA2/CA3 and OML of the dDG and the dotted line the separation between dDG and vDG. Scale bar: 500 μm. **c** High magnifications in the CA1 region (squares in **b**) stained for βgal. Scale bar: 100 μm. **d** High magnifications in the CA1 region of adult control and *BaxCKO* upon P73 staining and quantification (bottom panel) indicating an increase in P73⁺ cells in HF of mutant animals ((*p* = 0.0009 for HF, *p* = 0.9984 for vDG, two-way ANOVA, Sidak's multiple comparison test; *n* = 3 controls and *n* = 3 mutants). Scale bar: 100 μm. **e** High magnification of coronal sections of adult control and *BaxCKO* (all the samples are carrying the *R26^Tomato* reporter) hippocampus immunostained for Tomato (TOM), RELN and GABA. All TOM⁺ cells are RELN⁺ and GABA⁻ (*p* < 0.0001 for controls; *p* < 0.0001 for mutants) indicating they are CRs and confirming the increased number in HF, but not vDG in mutant animals (*p* = 0.0305 for HF, *p* = 0.2615 for vDG, two-way ANOVA, Sidak's multiple comparison test; *n* = 4 controls and *n* = 4 mutants). Scale bar: 100 μm. All data are presented as mean ± SEM. (*\*p* < 0.05; \*\**p* < 0.01; \*\*\**p* < 0.001; \*\*\*\**p* < 0.0001).

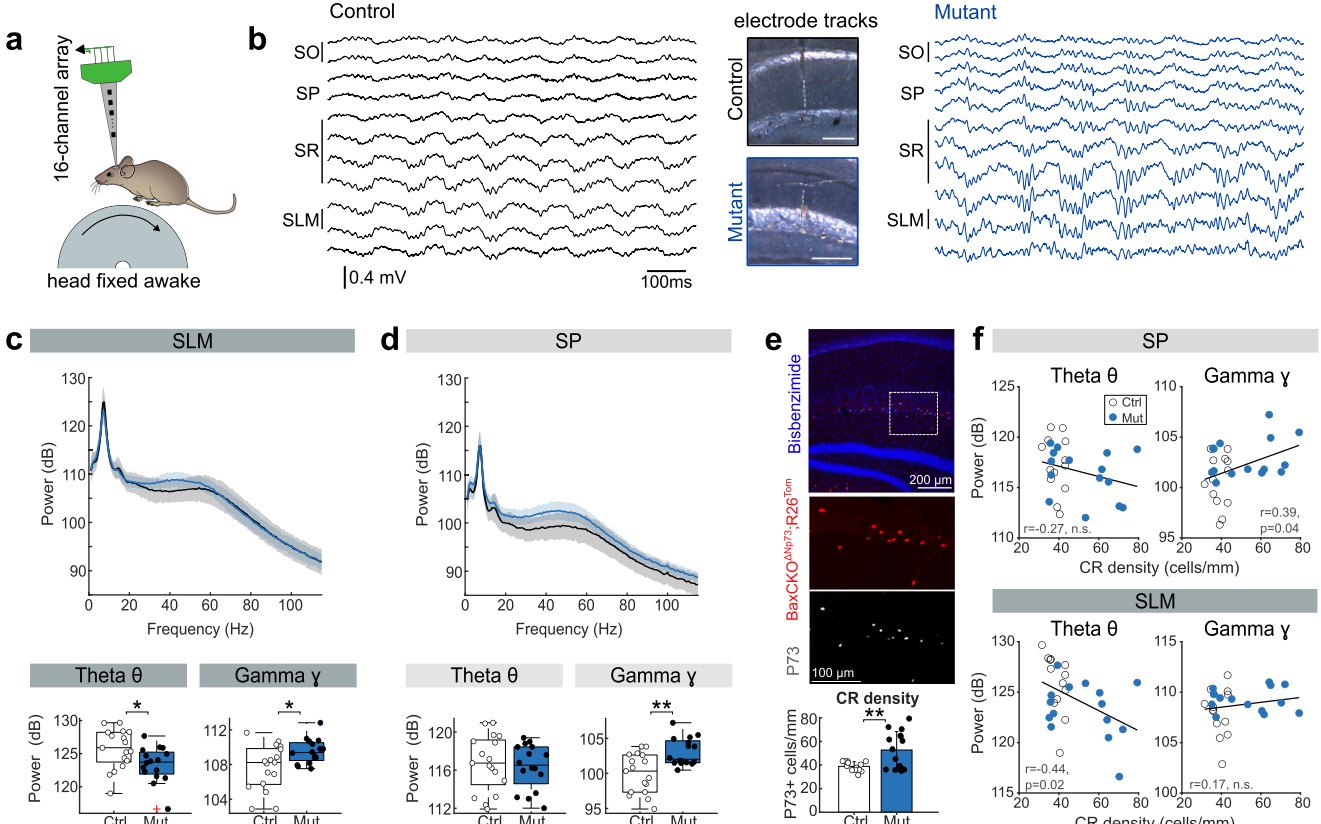

**Fig. 3 | *BaxCKO* mutants exhibit gamma rhythmopathies in the dorsal hippocampus. a** Experimental preparation. LFP signals were recorded with 16-channel linear silicon probes from the dorsal hippocampus of head-fixed mice (Mouse schema from Ann Kennedy. Mouse profile (Zenodo, 2020; doi:10.5281/zenodo.392591)). **b** Representative examples of LFP recordings from different layers in a control and a *BaxCKO* mutant. The probe position was validated (middle). SO stratus oriens, SP stratum pyramidale, SR stratum radiatum, SLM stratum lacunosum moleculare. **c** Boxplots showing distribution of theta and gamma power spectra recorded at SLM for controls (17 sessions from 3 mice) and *BaxCKO* mutants (16 sessions from 3 mice). The central mark is the median, the box edges are the 25th and 75th percentiles, and the whiskers are the data maxima and minima excluding outliers. Differences were observed in both theta ($p = 0.0211$, bottom left) and gamma ($p = 0.04$, Kruskal−Wallis test; bottom right). Red cross denotes an outlier. **d** Same as in **c** for SP. Group differences were observed for gamma power (40−80 Hz; $p = 0.0055$; Kruskal−Wallis test) but not for theta ($p = 0.614$). **e** Histological validation of enhanced CRs density from a given *BaxCKO* mouse section showing overlap of tdTomato reporter and the CR-specific P73 marker ($38.33 \pm 3.95$ cells/mm for controls and $52.76 \pm 15.66$ cells/mm for mutants, $p = 0.03$, Kruskal−Wallis. $n = 13$ controls and $n = 15$ mutants. Data presented as mean density ± SD. **f** Correlations between CR density at the HF within area CA1 and theta (left) and gamma (right) power recorded at the at SP (top panels) and SLM (bottom panels). Note correlation between CR density and gamma (Pearson correlation, $r = 0.39$, $p = 0.04$; two-tailed significance test), but not theta power at SP, and theta (Pearson correlation, $r = -0.44$, $p = 0.02$; two-tailed significance test), but not gamma, power at SLM. Data from 13 control sessions from 3 mice and 15 mutant sessions from 3 mice with clear probe track identification.

NPY$^+$ interneurons, in the CA1/CA2 hippocampal region of juvenile mutant animals.

### Glutamatergic neurons in the hippocampus show morphological alterations upon CR persistence

To further investigate the microcircuit defects suggested by in vivo recordings, we next questioned how CR persistence may affect the morphology and electrophysiological features of glutamatergic CA1 pyramidal neurons of adult mutant mice. Pyramidal cells were filled with biocytin in vitro using patch-clamp recordings followed by 3D reconstruction. Using Sholl analysis, we observed an increase in complexity (intersections) of apical dendrites (Fig. 5a), which was characterized by an increase in the total length, but not in the number, of dendritic branches (Figs. 5a and S5a, b). No change was detected in basal dendrites (Fig. S5a−c). Moreover, the density of spines on both basal and apical dendrites was enhanced (Fig. 5b) similarly to what was previously observed in the neocortex[29].

To investigate whether such morphological alterations were associated with functional defects, we recorded spontaneous excitatory and inhibitory post-synaptic currents (sEPSC and sIPSC, respectively) using in vitro patch-clamp recordings. No change was detected

in the amplitude or frequency of sEPSCs in mutant animals relative to controls (Fig. 5c). However, we did observe a significant increase in sIPSC frequency in mutant mice ($p = 0.0405$, Mann Whitney test) (Fig. 5d), consistent with enhanced gamma oscillations recorded in vivo. To further investigate the synaptic defects, we next evaluated spontaneous synaptic transmission independent of action potential initiation by recording miniature excitatory post-synaptic currents (mEPSCs) in CA1 pyramidal cells in the presence of tetrodotoxin (TTX 1 µM). The frequency and amplitude of somatically recorded mEPSCs were similar in control and mutant mice (Fig. S5d), suggesting no obvious differences are present between conditions in the total number of active spontaneous excitatory inputs onto CA1 pyramidal neurons. These results contrast with the observed increase in density of spines between WT and *BaxCKO* mutant mice (Fig. 5b). However, somatically recorded sEPSCs and mEPSCs mainly reflect synaptic activity from dendritic compartments close to the recording site (soma) and might not capture functional differences from distal parts of the dendritic tree. Interestingly, the difference in spine density between WT and *BaxCKO* mutant mice was particular evident in apical dendrites and less obvious in basal dendrites which could explain the lack of difference in somatically recorded sEPSCs and mEPSCs.

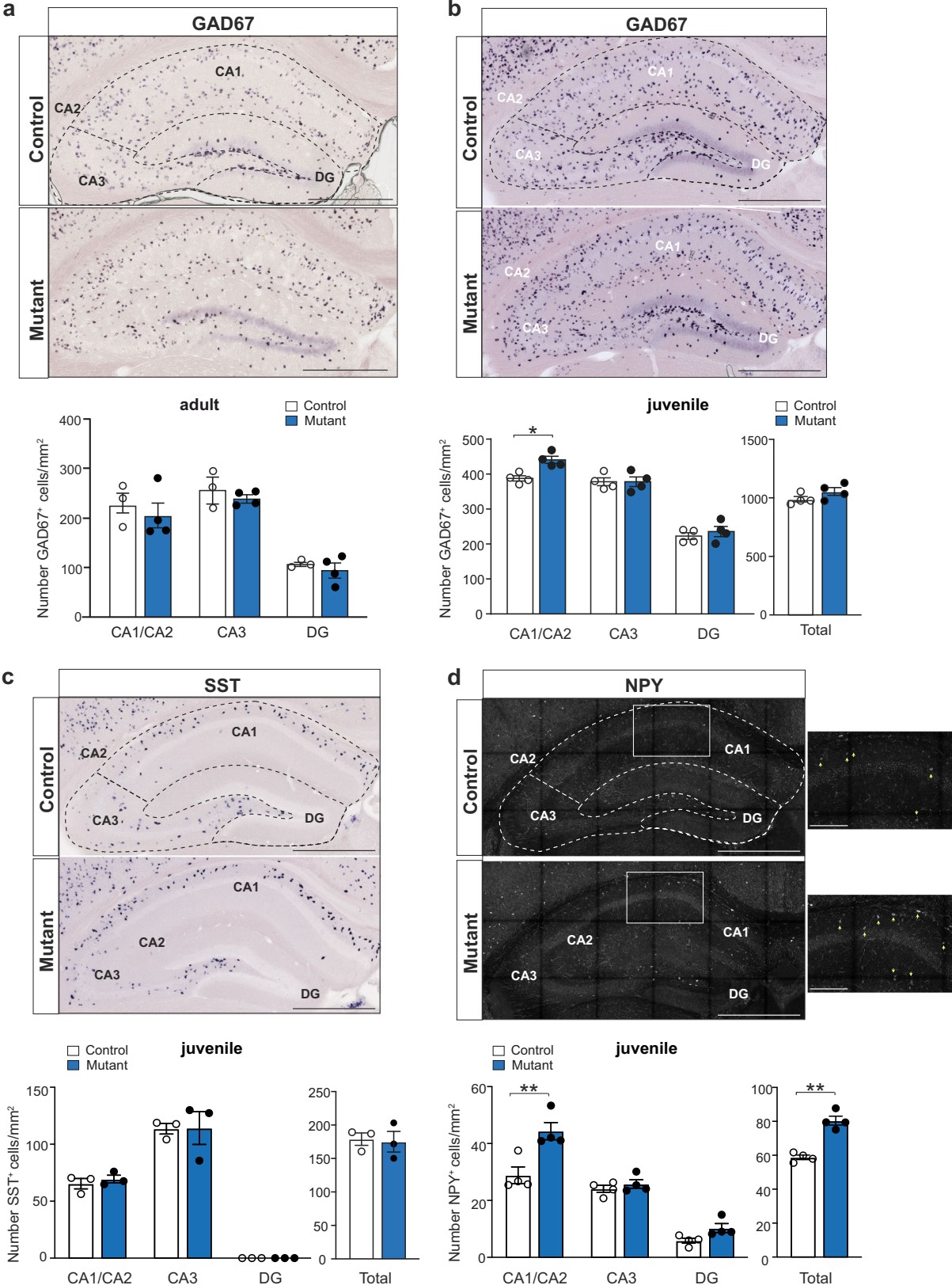

Moreover, the intrinsic electrophysiological properties and firing frequency vs. current injection (*F*−*I*) relationship of CA1 pyramidal neurons (estimated from a different set of experiments), did not show a significant difference between controls and mutants (Fig. S5e−h).

Altogether, our results demonstrate that aberrant survival of hippocampal CRs transiently alter the proportion of GABAergic interneurons and results in more frequent spontaneous inhibitory currents in vitro and theta-gamma rhythmopathies in vivo in adult *BaxCKO* mice, consistent with spatial memory deficits. These findings strongly suggest that PCD of CRs is important for proper maturation of neuronal networks, not only in neocortex, but also for the organization of hippocampal circuits.

**Fig. 4 | Enhanced survival of interneurons in the juvenile hippocampus. a** In situ hybridization on coronal sections of adult brains using a cRNA probe for GAD67 (GAD1) (top panels). Quantification of GAD67+ cells in the different hippocampal regions of control and mutant animals (bottom panel) showing no differences in the number of interneurons in distinct hippocampal regions ($p = 0.6947$ for CA1/CA2; $p = 0.8548$ for DG; $p = 0.9578$ for CA3, two-way ANOVA, Sidak's multiple comparison test, $n = 3$ controls, $n = 4$ mutants). Dotted lines in **a–d** represent the subdivisions in different regions: CA1/CA2, CA3 and DG. **b** In situ hybridization on coronal sections of juvenile brains using a cRNA probe for GAD67 (top panels). Quantification of GAD67+ cells in the different hippocampal regions of control and mutant animals (bottom panel) showing increased number of interneurons in CA1/CA2 ($p = 0.0118$), but not CA3, DG or total number of interneurons in mutant animals ($p > 0.9999$; 0.8129 and 0.3429, respectively; two-way ANOVA, Sidak's multiple comparison test; Mann–Whitney for total, $n = 4$ controls and $n = 4$ mutants). **c** In situ hybridization on coronal sections of juvenile brains using a cRNA probe for somatostatin (SST) (top panels). Quantification of SST+ cells of control and mutant animals (bottom panel) showing no differences in the number of interneurons in mutant animals ($n = 3$ controls and $n = 3$ mutants) in CA1/CA2, CA3 and DG or in their total numbers ($p = 0.9655$ for CA1/CA2; $p = 0.9998$ for DG; $p > 0.9999$ for CA3, two-way ANOVA, Sidak's multiple comparison test; $p > 0.9999$ for total, Mann–Whitney). **d** Immunofluorescence for NPY on coronal sections of juvenile brains (top panels). Quantification of NPY+ cells of control and mutant animals showing increased number of interneurons in CA1/CA2 ($p = 0.003$), but not CA3 or DG ($p = 0.9621$ and $p = 0.5889$ respectively, two-way ANOVA, Sidak's multiple comparison test), as well as an increase in the total number of NPY+ interneurons in mutant animals ($p = 0.0017$; Mann–Whitney; $n = 4$ controls and n = 4 mutants). High magnification of a CA1 portion (white square) (right panels) highlighting NPY+ neurons (yellow arrows). Scale bar: 100 μm. Scale bars in **a–d** low magnifications: 500 μm. Asterisks indicate statistical significance (*$p < 0.05$; **$p < 0.01$). All data are presented as mean ± SEM.

## CRs persistence leads to enhanced susceptibility to seizures

Although different studies associate both gain or loss of CRs with convulsive disorders, persistence of CRs has also been described in epilepsy-associated disorders in patients, based on histological studies[15–17]. Given the role of aberrant gamma rhythmopathies in epileptic phenotypes[44], we sought to investigate whether abnormal CR survival could also promote epileptiform activities. To this purpose, we first performed video-EEG recordings using wire recordings from the dorsal hippocampus and fronto-parietal cortex of juvenile freely moving mice, as previously described[46,47]. Hippocampal and cortical EEG activity was recorded for sessions of 3–4 h during days and nights over a period of 2 weeks. Analysis of the video-EEG recordings performed during the different sessions (total day shift = 18 h, total night shift = 12 h) did not reveal any major abnormal paroxysmal activity (Fig. S6a), consistent with the laminar LFP recordings reported above. We concluded that no major epileptiform activity and/or seizures could be spontaneously detected in juvenile *BaxCKO* mutant animals.

Many late-onset epileptic disorders result from a second hit insult during adulthood in the presence of underlying asymptomatic malformations[48]. We thus decided to exploit a pharmacological approach commonly used to induce epileptic seizures to determine whether CRs persistence may lead to predisposition to develop epileptic seizures. A single intraperitoneal injection of kainic acid (KA) at low dosage (15 mg/kg, i.p.) was performed in both male and female adult animals. We found that 94% of mutants ($n = 16$) developed a convulsive status epilepticus vs. only 26% in control adult males ($n = 23$) (Fig. 6a). Although the latency to develop seizures was the same for animals that underwent convulsions in both groups (responders) (Fig. 6c), the status epilepticus was longer in *BaxCKO* mutants compared to controls (Fig. 6a–c). Most animals recovered in the control group (4.35%), whereas 62.5% of mutants died within 20 min from seizure onset (Fig. 6a). Similar results were obtained in adult females with 87% of mutants ($n = 7$) developing seizures compared to 29% in controls (n = 7). Most mutant animals displayed faster and longer status epilepticus with a tendency ($p = 0.055$) for higher incidence of death (42.9% and 0% in controls) (Fig. S6b). Notably, when the same experiment was performed in juvenile mice, no differences in status incidence or mortality were observed (Figs. 6d and S6c), supporting the idea of a late-onset epileptic susceptibility in mutants. Taken together, these results show that the persistence of a small population of CRs may lead to epileptic prone phenotypes in adulthood in response to a second hit.

## Hyperexcitable entorhinal-hippocampal networks caused by CRs persistence

Given the increased susceptibility to KA-induced status epilepticus in *BaxCKO* mice, we then evaluated whether CR persistence altered patterns of neuronal network activity in temporal entorhinal-hippocampal circuits known to be responsible for the initiation of limbic-like

seizures[49,50]. To do so, we performed multi-electrode array (MEA) recordings in a pro-bursting ACSF (0 mM Mg$^{2+}$, 6 mM K$^+$) in entorhinal-hippocampal slices from control and mutant mice (Fig. 7a). While the individual burst frequency and duration was similar in slices from both genotypes, we observed that CR persistence caused a switch of the network activity pattern from bursting in controls to seizure-like events in mutants (Fig. 7b, c). The majority of control slices (85%; n = 17 out of 20 slices from 4 mice) displayed bursting activity, and seizures were observed in only 3 slices (seizure frequency: $0.29 \pm 0.05$/min; seizure duration: $33.5 \pm 3$ s; Fig. 7c, d). In contrast, such discharges were present in 55% of slices (n = 11 out of 20 slices from 4 animals) with a mean frequency of $0.41 \pm 0.08$/min and duration of $29.1 \pm 6.3$ s in mutant mice (Fig. 7c, d).

We also analyzed the spatial extension of the recorded epileptic-like discharges in control and mutant mice. To this end, we fragmented the entorhinal-hippocampal slices into sub-regions, namely, dentate gyrus (DG), CA3, CA1, subiculum (Sub), entorhinal (Ent) and perirhinal/ectorhinal (PRh/Ect) cortex, and evaluated the presence of the discharges in each sub-area (Fig. 7e). We found not only that there was a higher incidence of discharges in slices from *BaxCKO* mutants, but also that they spread more compared to control mice (Fig. 7e). Namely, in mutant mice the discharges involved the entire hippocampal formation and cortex (80% in DG and the totality of the seizures in all other sub-regions), while in control mice DG and CA3 were less involved (only 33% and 67% of epileptiform discharges).

Finally, we investigated seizure initiation sites in slices prepared from control and mutant mice by identifying the electrodes where seizures were initiated. We found that the proportion of seizures starting either in the hippocampus (DG, CA3, CA1 and Sub) or in the cortex (Ent and PRh/Ect) was similar in control and mutant mice (control: 54.5%/45.5%, $n = 11$ seizures form 3 slices; mutant: 56.8%/43.2%, $n = 37$ seizures from 11 slices; $p = 1.000$, Fisher's exact test, Fig. 7f, red line). However, by investigating the contribution of each subregion to seizure initiation, we found that while the Ent and PRh/Ect cortical regions were similarly involved in seizure initiation in both groups (control: 9.1%/36.4%, $n = 5$ seizures from 3 slices; mutant: 18.9%/24.3%, $n = 16$ seizures from 11 slices; $p = 0.6065$, Fisher's exact test; Fig. 7f), the hippocampal seizure initiation sites were differently distributed. Indeed, in mutant mice seizures starting in the hippocampal formation mainly originated in DG and CA3 subregions (24.3%/27%, $n = 19$ seizures from 11 slices), while in control mice 50% of the seizures ($n = 3$ seizures from 3 slices) originated in CA3, with no contribution from DG ($p = 0.035$, Chi-square test; Fig. 7f).

In all, these results indicate that aberrant CRs survival upon Bax inactivation not only increases network excitability, causing a switch from bursting to seizure activity, but also modifies seizure dynamics by changing seizure initiation sites and boosting seizure propagation. Thus, hyperexcitable temporal lobe networks, as tested in entorhinal-hippocampal slices, support the increased susceptibility of *BaxCKO*

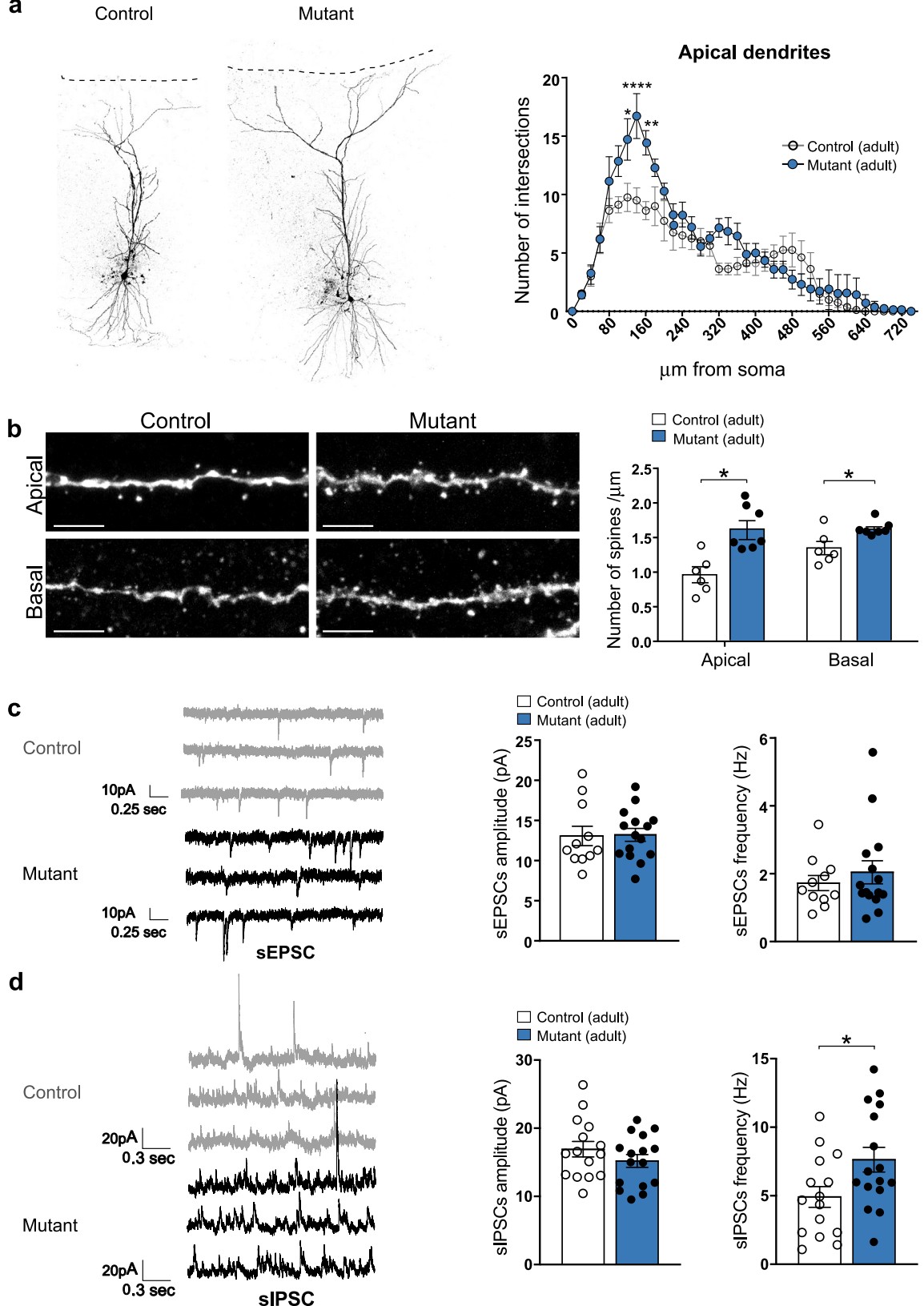

mice to develop lethal tonic-clonic status epilepticus upon kainate injection.

## Discussion

In the present study, we show that programmed cell death of hem-derived CRs is essential to achieve the correct maturation of hippocampal neuronal circuits and functions. While transient changes accompany the establishment of juvenile circuitry, the abnormal persistence of these CRs during advanced postnatal life upon Bax inactivation leads to impaired spatial and recognition memory functions, as well as susceptibility to temporal lobe epileptic seizures in adulthood. Our work provides a direct

**Fig. 5 | CRs abnormal survival leads to increased spine density and sIPSC frequency in adult CA1 pyramidal neurons. a** Representative examples of CA1 pyramidal neurons filled with biocytin in adult control and *BaxCKO* mutants (left panel). Sholl analysis for the apical dendrites in controls and mutants showing an increase of cell complexity between 120 and 160 μm from the soma ($p = 0.0133$, <0.0001 and 0.0012, respectively, post hoc Sidak's multiple comparison test, $F(1, 13) = 5.862$; $p = 0.0308$ for genotype, RM two-way ANOVA; $n = 8$ controls and $n = 7$ mutants; right panel). **b** Representative confocal images showing spines in apical (top) and basal (bottom) dendritic segments of CA1 pyramidal neurons in control littermates and *BaxCKO* adult mutants (left panels). Quantification of spine density (number of spines/μm right panels) in apical and basal dendrites showing an increase in mutant animals compared to controls ($p = 0.0260$ for apical and 0.0350

for basal; Mann–Whitney test, $n = 6$ controls and $n = 6$ mutants for apical and $n = 6$ controls and $n = 7$ mutants for basal). **c** Representative traces of sEPSCs recorded from CA1 pyramidal neurons in voltage-clamp mode at −60 mV in brain slices from adult control and mutant animals (left). No changes in mean sEPSC amplitude ($p = 0.6461$) or frequency ($p = 0.6832$) between mutant and control animals ($n = 11$ controls and $n = 15$ mutants, Mann–Whitney test). **d** Same as C but for sIPSCs recorded in voltage-clamp at 0 mV. sIPSC frequency was significantly increased ($p = 0.0405$) in brain slices from mutant animals while sIPSCs amplitude remained constant ($p = 0.2995$; Mann–Whitney test, $n = 15$ controls and 16 mutants). Scale bar: 5 μm. Asterisks indicate statistical significance (*$p < 0.05$; **$p < 0.01$; ****$p < 0.0001$). All data are presented as mean ± SEM.

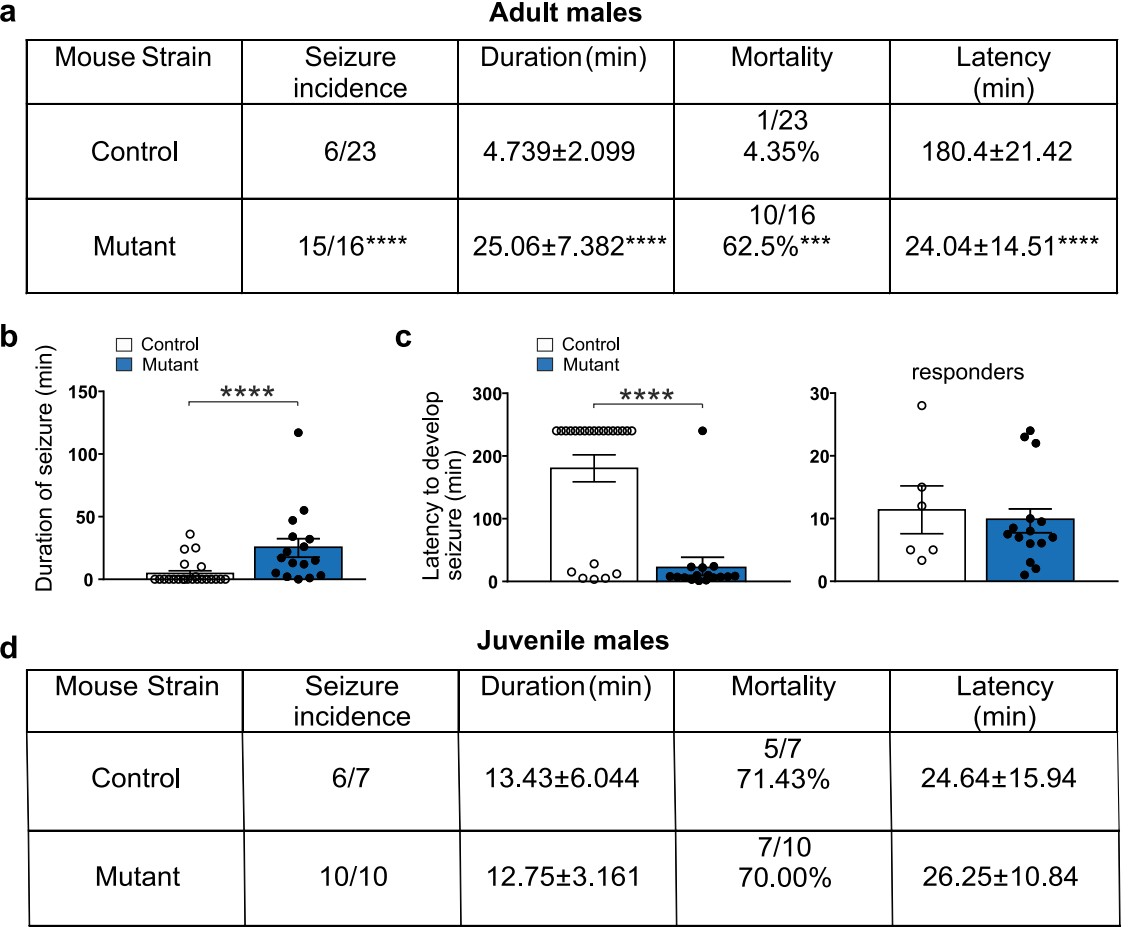

### a
### Adult males

| Mouse Strain | Seizure incidence | Duration (min) | Mortality | Latency (min) |
|---|---|---|---|---|
| Control | 6/23 | 4.739±2.099 | 1/23 4.35% | 180.4±21.42 |
| Mutant | 15/16**** | 25.06±7.382**** | 10/16 62.5%*** | 24.04±14.51**** |

### b
### c

responders

### d
### Juvenile males

| Mouse Strain | Seizure incidence | Duration (min) | Mortality | Latency (min) |
|---|---|---|---|---|
| Control | 6/7 | 13.43±6.044 | 5/7 71.43% | 24.64±15.94 |
| Mutant | 10/10 | 12.75±3.161 | 7/10 70.00% | 26.25±10.84 |

**Fig. 6 | Increased susceptibility to develop tonic-clonic seizures leading to death upon KA injection in adult male mutant mice. a** Enhanced seizure incidence, duration, mortality and latency upon KA injection (15 mg/kg, i.p.) in adult mutant males compared to control littermates ($p < 0.0001$; $p < 0.0001$; $p = 0.0001$ and $p < 0.0001$, respectively; Mann–Whitney test, $n = 23$ controls and $n = 16$ mutants). Most control animals did not develop seizures. **b** Increased duration of seizures in adult mutant animals compared to controls ($p < 0.0001$; Mann–Whitney test, $n = 23$ controls and $n = 16$ mutants). **c** Decreased latency to develop seizures in adult mutants compared to controls (left panel, $p < 0.0001$, Mann–Whitney test, $n = 23$ controls and 16 mutants), indicating that more mutant animals were

developing seizures. Nevertheless, taking into consideration only the animals in both groups (controls and mutants) which developed seizures (responders: 11.4 ± 3.82 for controls and 9.64 ± 1.91 for mutants), no differences were detected in the latency to develop seizures (right panel, $p = 0.8348$, Mann–Whitney test, $n = 6$ controls and $n = 15$ mutants). **d** Unchanged seizure incidence, duration and mortality upon KA injection (15 mg/kg, i.p.) in juvenile (P24) mutant males compared to control littermates ($p = 0.4118$ for seizure incidence; $p = 0.7611$ for duration; $p > 0.9999$ for mortality and $p = 0.5503$ for latency, Mann–Whitney, $n = 7$ controls, $n = 10$ mutants). Asterisks indicate statistical significance (***$p < 0.001$; ****$p < 0.0001$). Values are expressed as mean ± SEM.

demonstration of the functional consequences of CR persistence in mature brain function.

CRs are generated mainly at three different sites of the developing pallium, septum, hem and the pallial-subpallial boundary. Hippocampal CRs, which are mostly generated in the cortical hem[28,39], display unique features. While most neocortical CRs (97–98%) die during the first two postnatal weeks[3,13,51], hippocampal CRs remain longer all along the fissure of juvenile mice[27,28,39,52].

Moreover, genetic tracing studies are now providing compelling evidence that about 30% of CRs survive to adulthood in the mouse hippocampus[27,28]. Interestingly, the density of CRs in the adult hippocampus is similar to that of the immature neocortex, suggesting that neocortical and hippocampal circuits exhibit distinct tolerance thresholds for CRs activity.

Are persistent CRs deleterious for hippocampal function? A decrease in hippocampal CR number with age was described[27,39],

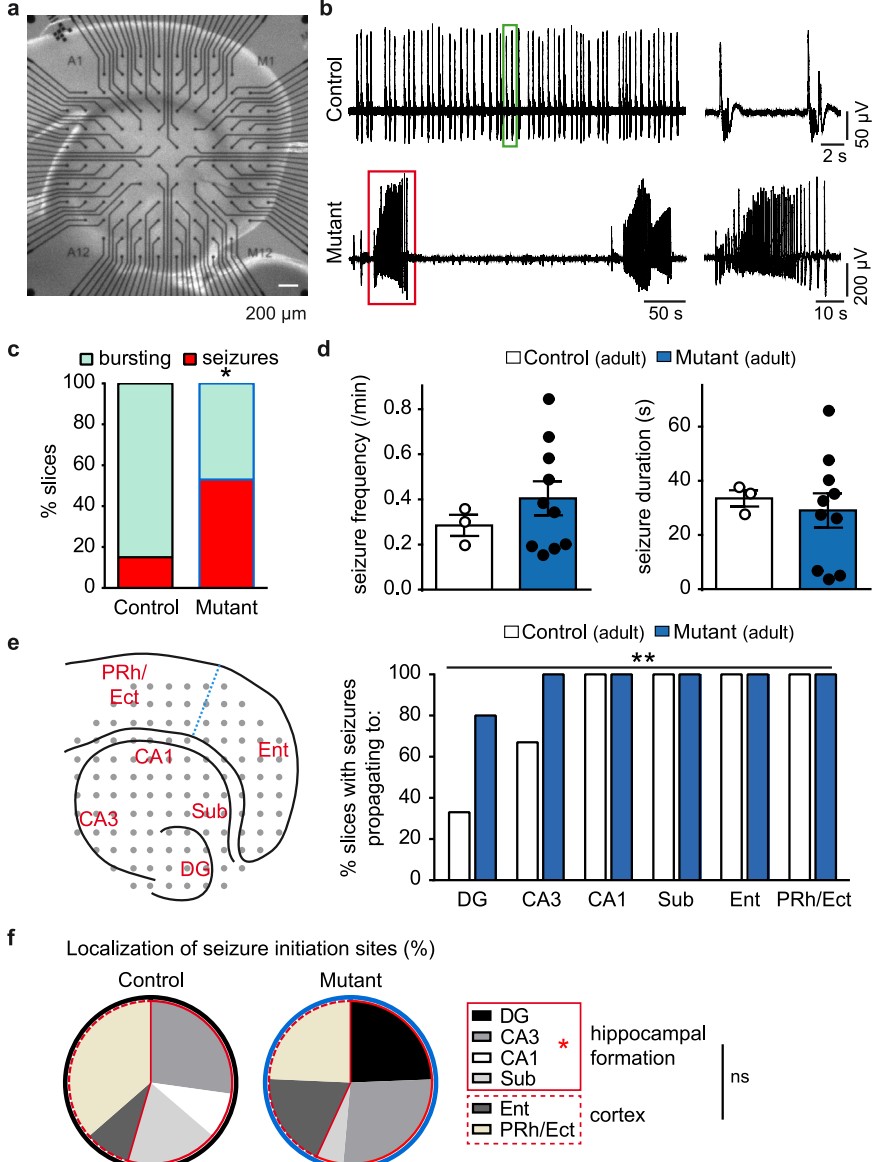

**Fig. 7 | Survival of CRs induces a switch of the activity pattern recorded in entorhinal-hippocampal slices from bursting to seizures. a** Picture of a cortico-hippocampal mouse slice in a multi-electrode array (MEA) chamber. Scale bar: 200 μm. **b** On the left, representative MEA recordings of bursting and seizure-like activity in 0Mg-6K ACSF in control (top) and mutant (bottom) adult mice. The events highlighted by green and red rectangles are zoomed on the right. **c** Quantification of the percentage of slices with bursting (green) or seizure (red) activity in control and mutant mice (*n* = 20 slices from 4 mice for both genotypes; *p* = 0.0187, Fisher's exact test). **d** Quantification of seizure frequency and duration in slices displaying seizures in control (n = 3 slices) and mutant (*n* = 11 slices) mice.

**e** Left, schematic representation of a cortico-hippocampal slice with sub-regions: DG, CA3, CA1, subiculum (Sub), entorhinal (Ent) and Perirhinal/Ectorhinal (PRh/Ect) cortex. Right, quantification of the percentage of slices with seizures propagating to the different subregions of entorhinal-hippocampal slices (*p* = 0.0011, Chi-square test). **f** Quantification of seizure initiation sites (%) in cortico-hippocampal slices from control and mutant mice (*p* = 1.000 (cortex vs hippocampal formation); *p* = 0.6065 (Ent vs PRh/Ect); *p* = 0.035 (hippocampal subregions), Fisher's exact test and Chi-square test). Asterisks indicate statistical significance (**p* < 0.05; ***p* < 0.01). Values are expressed as mean ± SEM.

suggesting that their removal is important for the maturation of functional microcircuits. Up to now, alterations in pyramidal cell morphology and physiology were detected in the developing somatosensory cortex upon increased or decreased CR numbers, indicating that they regulate spine density and excitability of cortical circuits[26,29,53]. Furthermore, defects in anatomical properties of the adult hippocampus were observed in animal models with enhanced CR numbers, such as in the *reeler* mouse mutant[54], upon progesterone receptor antagonist treatment[55] or enrichment[28]. While a lack of *reelin* was associated with human TLE[56] and seizure susceptibility following intra-hippocampal injection of kainate in mice[57], no direct evidence has been provided so far for a leading role of abnormal CR persistence. Yet,

persistence of CRs during advanced postnatal life is reported in several pro-epileptogenic pathologies[14–17,58]. Our *BaxCKO* model represents a mouse model in which Bax invalidation is specifically induced in septum- and hem-derived CRs, allowing assessment of the morphological and functional impact of CRs persistence on adult brain structure and cognition.

We found both transient and permanent changes of hippocampal cellular and microcircuit function upon persistence of hem-derived CRs in *BaxCKO* mice. During juvenile age, abnormal accumulation of hippocampal CRs concurred with transient outnumbering of NPY+ interneurons, which returned to control levels in adulthood. In control adults, CRs are embedded in local networks where they receive

GABAergic input from NPY+ and OLM SST+ interneurons[45]. As around 30–40% of GABAergic interneurons die at early postnatal age[3], our results strongly suggest that the survival of their CR targets leads to a concomitant transient increased survival of NPY+, but not SST+, interneurons and, thus, that CRs death contributes to the regulation of inhibition in the hippocampal CA1 region. Additionally, adult *BaxCKO* mutant CA1 pyramidal neurons display altered morphology with an increased number of dendritic spines. During electrophysiological investigation, we found an increased frequency of sIPSCs and a surprisingly equal number of sEPSCs and mEPSCs events in adult pyramidal neurons upon CRs survival. However, in our anatomical quantifications, the most conspicuous morphological differences in CA1 neurons were observed in apical dendrites. Thus, the apparent discrepancy between estimated number of spines and sEPSCs (or mEPSCs) frequency could be explained by the preferential detection of synaptic events from somatic and proximal dendritic compartments. However, we cannot exclude the possibility that the additional spines present in CA1 neurons from mutant mice are not equipped with AMPA receptors (silent synapses) and hence do not contribute to EPSCs recorded at −60 mV. Together, these findings suggest that balanced CR-driven excitatory and GABAergic inhibitory activity is required to ensure proper functional networks in adults and that a transient defect caused by their aberrant persistence in adolescence might permanently influence circuit operation later in life. Indeed, both in the wildtype neocortex and hippocampus, CRs demise correlates with circuit maturation. At juvenile stages, neuronal networks maturation and spine pruning are in progress[59,60]. Thus overall, our and others' data are consistent with changes in CRs number modifying the timing of network development. Our data on transient defects in interneuron numbers exclusively in juveniles are consistent with a possible delay in interneuron death, synaptic pruning and overall in cortical development.

Nevertheless, we show that animal behaviors are specifically altered in adults and CRs survival persists in CA1. Importantly, we found that *BaxCKO* mice exhibit severe learning and memory impairments at adult age. Although we cannot conclude that all behavioral defects observed in these mice are solely hippocampal-driven, the use of behavioral tests assessing selective hippocampal-dependent spatial learning and memory (i.e. NOL and MWM tests) together with the rhythmopathies and MEA alteration, strongly argue in favor of an important contribution of alterations of hippocampal circuits in our phenotype. Moreover, the presence of CRs do not alter all circuits and behavioral functions related to the hippocampus. Indeed, by contrast to the spatial learning and memory, we did not observe any contextual fear memory deficits in *BaxCKO* mice. The encoding conditioned fear memory relies on selective hippocampal–amygdala pathways, which seem not to be disturbed by the persistence of CRs. Moreover, we did not observe any anxiety- and depression-like behavioral phenotype in the *BaxCKO* mice. Interestingly, recent studies[61] showed that glutamate synaptically released by CRs is critical for the regulation of hippocampal microcircuits and the control of learning and memory functions. Similarly to our mouse model, they have shown impairments of spatial learning and memory, however they did not observe significant changes in recognition memory. Taken together, these results strongly support the importance of CRs for the regulation of hippocampal circuits. The differences in recognition behavior between both mouse models also suggest the existence of additional players than glutamate release mediating the effects observed upon CR survival in the *BaxCKO* mice and/or region-specificity of functional defects caused by hippocampal CR persistence.

Consistent with increased sIPSC activity recorded in adult *BaxCKO* mice, we found enhanced gamma activity in the dorsal hippocampus of these mice recorded in vivo. These changes were more prominent at the SP of mutants. In addition, the amplitude of theta oscillations was significantly attenuated at the SLM. Both theta and gamma hippocampal oscillations result from the balanced activity of local GABAergic interneuron subpopulations[62,63]. A minority of hippocampal CRs that survive in adulthood have been shown to integrate into the local microcircuitry, supporting their involvement in spatial memory[27,39]. Thus, the observed changes in hippocampal oscillations may result from the maladaptive surplus of CRs found in the *BaxCKO* mutants. In accordance with these observations, we found a correlation between CR density at the HF and increased gamma power at the SP, as well as with the attenuation of power in the theta band at the SLM. A proper tuning of oscillations is critical for hippocampal memory function and so the existence of theta–gamma rhythmopathies may have deleterious effects on hippocampal-dependent cognition, consistently with the behavioral defects in *BaxCKO* mice.

CRs localized in the SLM and OML were shown to be an important temporary target for entorhino-hippocampal axons[24,27,39,45,53,64–66] and CR ablation indeed determines an off-targeting of these afferent inputs[24]. Given the presence of an enhanced number of CRs in the HF region of mutant *BaxCKO* mice, the possibility arises that their persistence could interfere with entorhino-hippocampal communication, which plays major roles in memory function. Our data is in agreement with a deleterious effect of CR persistence in spatial memory function, but not in aversive memory function that rely on DG and CA3 circuits. Consistently, lesions of the medial entorhinal cortex have been shown to lead to defects in spatial memory in the Morris water maze, but not in paradigms such as novel object recognition and contextual fear conditioning[67,68]. Together, these results strongly argue against a simple effect of CRs and interneuron survival in CA1 on the entorhinal-hippocampal system as mediating the behavioral defects observed in *BaxCKO* mutants.

Importantly, the entorhinal-hippocampal network is characteristically involved in limbic epileptic disorders[49,50]. However, we found no spontaneous epileptic phenotype in *BaxCKO* mice, but an increased susceptibility to pro-epileptogenic insults, consistent with many late-onset epileptic disorders resulting from a second hit in otherwise asymptomatic malformations[48]. This may be linked with enhanced gamma oscillations recorded in *BaxCKO* mice, since synchronous network inhibition was shown to play a pivotal role in the initiation of limbic focal seizures[21]. In line with this, we found signs of hyperexcitability, which manifested as a switch of the activity pattern from bursting to seizures in combined entorhinal-hippocampal slices prepared from *BaxCKO* mice. Taken together, the altered morphology of CA1 pyramidal neurons, the increased frequency of sIPSCs and the transient survival of hippocampal CA1 interneurons could drive an electrical alteration of hippocampal circuits, predisposing mutant animals to develop lethal epileptic crises.

Our results point to the type of adult onset epilepsy caused by persistence of CRs as a "two hit" process. The "two-hit" hypothesis in late-onset epilepsy was put forward for hippocampal sclerosis and mesial TLE[69]. Accordingly, persistence of CRs was detected in pathological conditions associated with epilepsy, including sclerosis and TLE[17,70], corroborating our findings that these neurons could indeed be involved in the insurgence of seizures. Second possible hits in humans such as genetic background, brain lesions, dysplasia or tumors, encephalitis, fevers, but also depression, stress or sleep apnea were proposed[69]. For instance, gender specificity of the glucocorticoid response to early life stress may be involved in the occurrence of spontaneous seizures only in adult males and not in females[71]. In fact, we found that epileptic responses to KA injections were more pronounced in adult mutant males.

Finally, while several studies support the idea that Reln insufficiency leads pro-epileptogenic changes[56,57,72–75], we found no

evidence of either an increase or decrease of Reln levels in the *BaxCKO* mutant hippocampus. Interestingly, recent data demonstrate that CRs also release glutamate contributing to important microcircuit function and behavior[61]. In this model glutamate release from CRs is inhibited, suggesting that similar to Reln levels, both the increase and decrease of CR glumatamatergic contribution may lead to circuit alterations. Possibly, the pro-epileptogenic effects in response to second hit insults in *BaxCKO* mice may reflect the enhanced active microcircuit contribution of persistent CRs[76]. Future studies will be required to tease out the specific roles of CR-mediated Reln and glutamatergic release in hippocampal function and dysfunction.

Altogether, our results indicate that the aberrant survival of even a small proportion of usually transient CRs from juvenile age to adulthood may have severe deleterious effects on brain structure and cognitive functions. This study also reinforces the notion that PCD of transient cell types is essential for the formation, maturation and correct functioning of neuronal circuits. Undetected or compensated defects in PCD during early postnatal cortical development could represent a yet underestimated cause of late-onset epilepsy. This work may thus provide the foundation for novel therapeutic avenues to treat a wider spectrum of neurodevelopmental and neuropsychiatric disorders.

## Methods

### Animals

$\Delta Np73^{CreIRESGFP}(\Delta Np73^{Cre})$[77], $Tau^{loxP\text{-}stop\text{-}loxP\text{-}MARCKSeGFP\text{-}IRES\text{-}nlslacZ}$ $(Tau^{GFP})$[78] and $ROSA26^{loxP\text{-}stop\text{-}loxP\text{-}Tomato}$ $(R26^{Tom})$[79] transgenic mice were kept in a C57BL/6 J background. The $Bax^{tm2Sjk;Bak1tm1Thsn}$/J line[80] harboring the *floxed Bax* and the *Bak* knock-out alleles was purchased from the Jackson laboratory as mixed B6;129. The $\Delta Np73^{Cre}$ line was crossed with the $Bax^{tm2Sjk;Bak1tm1Thsn}$/J line $(Bax^{lox/lox})$ to inactivate Bax function in specific CR subtypes and with the $Tau^{GFP}$ and $R26^{Tom}$ reporter lines to permanently label CR subtypes. For histological analysis $\Delta Np73^{Cre+/-};Bax^{lox/+}$ animals were used as controls, while for behavioral and electrophysiological analysis $\Delta Np73^{Cre+/+};Bax^{lox/lox}$ were used as control littermates. $\Delta Np73^{Cre+/-};Bax^{lox/lox}$ animals (*BaxCKO*) were used as mutants all along the study. Experiments were performed in both males and females aged juvenile (3–5 postnatal weeks), young adult (6–9 postnatal weeks) and adult mice (>3 months old).

### Ethics declaration

For experiments in Paris, all animals were handled in strict accordance with good animal practice as defined by the national animal welfare bodies, and all mouse work was approved by the French Ministry of Higher Education, Research and Innovation as well as the Animal Experimentation Ethical Committee of Paris Descartes University (CEEA-34, licence numbers: 18011-201801261202754 No. 2018121415089383−V8 APAFiS # 23831). Animals were kept in ventilated cages at a temperature of 20 °C and humidity of 50%. For in vivo experiments in Madrid, all protocols and procedures were performed according to the Spanish legislation (R.D. 1201/2005 and L.32/2007) and the European Communities Council Directive 2003 (2003/65/CE) for animal research. Experiments were approved by the Ethics Committee of the Instituto Cajal and the Spanish Research Council (PROEX162-19).

### Behavioral tests

All mice were under constant environmental enrichment and were handled for 3–5 days for at least 3 min before the beginning of the behavioral tasks. For each test, mice were transported a short distance from the holding mouse facility to the testing room in their home cages. The tests were performed by an experimentalist blind to the genotypes or treatment of the mice under study.

### Juveniles (3–5 weeks old) and young adults (6–9 weeks old) animals

Primary screen: 3–5 weeks (Males (M) and Females (F)) (cohort1)
Open-field: 5–6 weeks (M) (cohort 2)
3-chamber test: 6 weeks (M) (cohort 2)
FST: 7 weeks (M) (cohort 2)
Rotarod: 8 weeks (M) (cohort 2)
Cookie test: 9 weeks (M) (cohort 2)
EPM: 6 weeks (M) (cohort 3)
Y-maze: 7 weeks (M) (cohort 3)

The different tests were performed on different cohorts of mice.

**Primary screen.** Animals (both males and females) were tested in the primary screen at 3–5 weeks of age as previously described[30,81].

**Cookie test.** Mice were housed singly for three days before the test and were fastened 24 h prior to the test. The day of the test one mouse is placed alone in a Plexiglas cage (35 × 20 × 15 cm) with 3 cm bedding. A palatable food (cookie) is hidden under the bedding. Testing was conducted over 3 days. On day 1, habituation trial, mice have to find a piece of food (olfaction) placed on top of the bedding in the center of the cage. One minute after having eaten the piece of food (motivation), mice are returned to their home cage. On days 2 and 3, the experimental sessions consisted of two trials separated by a 2 ± 0.5 min inter-trial period in the home cage. At the beginning of each trial, a piece of food was buried beneath 3 cm of bedding in a pseudo-randomly selected location. The time taken for the mouse to find the buried piece of food was recorded and the mouse returned to the home cage 1 min after having eaten it. The latency to find the piece of food is taken as an index of olfaction. An animal which failed to find the food within 10 min was removed from the cage and a score of 600 s was assigned.

**Y-maze.** The Y-maze was made of three arms set at angles of 120°. It was surrounded by numerous objects which served as visual extra-maze cues. The animal was placed at the center of the maze and allowed to move freely for a 10-min session. The number of entries in the different arms as well as the percentage of alternation were recorded to assess spatial working memory. An alternation is defined as successive entries into the three arms on overlapping triplet sets.

**Open field (OF).** Activity was measured in open-fields (actimeters) consisting in Plexiglas transparent open-boxes (40 cm L, 40 cm W, 40 cm H) during 1 h. The distance traveled (horizontal activity) was recorded with infrared photobeams detection systems (Acti-track, LSI Leticca, Panlab). Actimeters were placed in compartments (one compartment for each actimeter), under a dim light (20 Lux in the middle of the box) so as each open-field was visually isolated from the others and from the experimental room. There was no other light in the room during experiments.

**Elevated plus maze (EPM).** The experiment was conducted in a plus-shaped maze, constructed from gray Perspex, elevated 50 cm from the floor, with two opposite open arms, 27 × 5 cm, crossed at right angles by two arms of the same dimensions but enclosed by 15 cm high opaque walls with an open roof. In addition, a 0.2 cm high edge surrounds the open arms to avoid falls. Light intensity was 5 Lux at the level of open arms. Animals' behavior was recorded by a camcorder placed 1.5 m above the maze. The time spent into the arms as well as the entries were automatically measured using Videotrack v2.6 Automated Behavioural Analysis (ViewPoint) tracking software. Animals were individually placed for a 5-min session in the center of the plus-maze facing one open arm. The data recorded were the

percentage of entries in open arms (100 × number of entries in open arms/number of entries in open + enclosed arms), the percentage of time spent in open arms (100 × time spent in open arms/time spent in open + enclosed arms), and the total number of arm entries. Animals that fell from the maze were removed from the experiment.

**3-chamber sociability test**. This test assesses sociability in mice as they normally prefer to spend more time with another mouse then in an empty chamber. The test was carried out in 3-chamber boxes in Plexiglas (60 cm long, 40 cm large, 35 cm high). Boxes were divided into three compartments (chambers) of the same dimension by walls with small square openings (5 × 3 cm) allowing access into each chamber. Two cylindrical acrylic cages (20 cm high, 10 cm diameter) in which an object or a mouse can be enclosed, were placed in a corner of each side chambers. Two cubes (4 × 4 × 4 cm; dark colored) were used as objects. Extra mice, used as stimulus mice, were C57BL/6 J mice of the same gender and of the same age. A video camera located at 1.5 m above the 3-chamber boxes was connected to an image analysis system (ViewPoint, France), allowing to record the distance traveled and the time spent by animals in the apparatus. The subject was first placed in center chamber for 5 min, without access to side chambers. Then, two empty cylinders were placed in a corner of each side chamber. Doors were removed and the subject was allowed to explore the three chambers for a 10 min habituation session. The animal was then confined in the center chamber. A stimulus mouse was placed in the cage located in one side chamber, and an object was placed in the cage located in the other side chamber. The side of object and stimulus mouse was pseudo-randomized in way that within all groups, the stimulus mouse was placed in one side chamber for half of animals and in the other side chamber for the other half. Doors were opened and the subject was allowed to explore all arenas for a 10 min session. Data recorded were the time spent exploring the box containing the stimulus mouse and the time spent exploring the object.

**Forced swim test (FST)**. This test is based on the observation that rodents, after initial escape-oriented movements, develop an immobile posture when placed in an inescapable situation. Each mouse is placed in a cylinder (height: 25 cm and diameter: 10 cm) filled with water (23–25 °C). Mice were tested for 6 min and the time spent immobile (behavioral despair) or climbing was quantified.

**Rotarod**. Balance and motor coordination were tested using an accelerating rotarod (Ugo Basile, Bioseb).

The animal was placed on the rod which rotated at 3 round per min (RPM) for 1 min. An animal which fell from the rod during this period was placed again on the rod. Then, the speed of the rod progressively accelerated from 3 to 32 RPM during 9 min. The animal was placed again on the rod after the first two falls and was removed from the apparatus after the third fall.

Animals were subjected to two sessions 24 h apart. The following parameters were recorded for each session: the number of falls during the first 1-min period, the number of falls during the following 9-min period, the first, second and third fall latency (min: 60 s–max: 600 s if no falls occurred) and the mean fall latency.

**Adults (3-month-old animals)**. Mice were transported from the holding mouse facility to the testing room in their home cages and left undisturbed for at least one hour before the beginning of the test. Two independent cohorts of mice were used for each behavioral tests and in both male and female adult mice. One week elapsed between two tests. The sequence of behavioral analyses was in order: *Open Field* (OF), *Dark and Light transition* (D/LT), *Novel object recognition* (NOR), *Novel object location* (NOL) and either *Morris Water Maze* (MWM) or *Contextual fear conditioning* (CFC) tests. The MWM was never

performed after CFC and inversely. Consequently, the CFC or MWM tests represent an ending procedure.

**Open field test (OF)**. This test takes advantage of the aversion of rodents to brightly lit areas. Each mouse is placed in the center of the OFT chamber (43 × 43 cm chamber) and allowed to explore for 30 min. Mice were monitored throughout each test session by infrared light beam activity monitor using actiMot2 Software (PhenoMaster Software, TSE). The overall motor activity was quantified as the total distance traveled (ambulation). Anxiety was quantified by measuring the time and distance spent in the center versus periphery of the open-field chamber.

**Light-to-dark transition test (D/LT)**. This test is based on the innate aversion of rodents to brightly illuminated areas and on their spontaneous exploratory behavior in response to the stressor that light represents. The test apparatus consists of a dark, safe compartment and an illuminated, aversive one. The lit compartment was brightly illuminated with an 8 W fluorescent tube (1000 lx). Naive mice were placed individually in the testing chamber in the middle of the dark area facing away from the doorway to the light compartment. Mice were tested for 10 min, and two parameters were recorded: time spent in the lit compartment and the number of transitions between compartments, indices of anxiety-related behavior and exploratory activity. Behavior was scored using an infrared light beam activity monitor using actiMot2 Software (PhenoMaster Software, TSE) and it was statistically analyzed using Prism program.

**Novel object recognition test (NOR)**. We used a modified version of the NOR test described by Glatigny et al.[31]. The testing room was lit with two 60 W light bulbs and test sessions were recorded with a camera placed above the test arena (a gray plastic box, 60 × 40 × 32 cm). The light intensity was equal in all parts of the arena (approximately 20 lx). Two different objects were used: a blue ceramic pot (diameter: 6.5 cm; maximum height: 7.5 cm) and a clear, plastic funnel (diameter: 8.5 cm; maximum height: 8.5 cm). The object that serves as the novel object and the left or right placement of the objects were counterbalanced within each group.

The NOR paradigm consists of three phases (over 3 days): a habituation phase, a training phase, and a testing phase. Mice were always placed in the center of the arena at the start of each exposure. On day 1 (the habituation phase), mice were given 5 min to explore the object-free arena and were then returned to their home cage. On day 2 (the training phase), mice were allowed to explore two identical objects (arranged in a symmetric opposite position from the center of the arena) for 10 min and were then returned to their home cage. On day 3 (the testing phase), mice were given 15 min to explore a familiar object and a novel one in the same arena and in the same positions as in the training phase. The following behaviors were scored as exploration: sniffing, licking, touching the object with the nose or with the front legs, or proximity (≤1 cm) between the nose and the object. If the mouse was on top of the object or completely immobile, exploration was not scored. The preference index for the novel object was calculated as (time spent exploring the new object/the total time spent exploring both objects), and the discrimination index was calculated as (time spent exploring the new object – time spent exploring the familiar object)/(total time spent exploring both objects). Behavior was scored on video by two observers blinded to the mice's treatment.

**Novel object location test (NOL)**. In the novel object location task, all procedures were identical to the NOR task except that mice encountered two familiar objects (rather than one familiar object and one novel object) during the testing phase. However, one of the familiar

object was located in a new position in the arena. The duration and frequency of exploration of the relocated object constitute indices of memory performance. Behavior was scored on video by two observers blinded to the mice's treatment.

**Morris water maze (MWM).** The MWM with an automatic tracking system was employed for assessing spatial learning and memory. The apparatus was a white circular swimming pool (diameter: 200 cm, walls: 60 cm high), which was located in a room with various distal cues. The pool was filled with water (depth: 50 cm) maintained at $22\,°C \pm 1\,°C$, which was made opaque by the addition of a nontoxic white paint. A 12 cm round platform was hidden 1.0 cm below the water surface. The maze was virtually divided into four arbitrary, equally spaced quadrants delineated by the cardinal points north (N), east (E), south (S), and west (W). The pool was located in a brightly lit room. Extra maze geometric and high-contrast cues were mounted on the walls of the swimming pool with the ceiling providing illumination. Each daily trial consisted of four swimming trials, in which each mouse was placed in the pool facing the wall of the tank and allowing the animal to swim to the platform before 120 s had elapsed. A trial terminated when the animal reached the platform, where it remained for 5 s. Mice were removed and placed back in their home cages for a 5 min inter-trial interval. To prevent hypothermia, the animals were gently dried with a paper towel between and after the trials. The starting point differed at each trial, and different sequences of release points were used from day to day. Swimming time to the platform was calculated as an evaluation of performance of the mice to locate the target.

**3-foot shock contextual fear conditioning (CFC).** Mice were tested individually inside the conditioning chambers (Bioseb (France); $25 \times 25 \times 25$ cm) located inside a larger, insulated plastic cabinet that provided protection from outside light and noise ($67 \times 55 \times 50$ cm). Floors of the chamber consisted of 27 stainless steel bars wired to a shock generator with scrambler for the delivery of foot shock. Signal generated by the mice movements was recorded and analyzed through a high sensitivity weight transducer system and analysis of time active/time immobile (Freezing) was performed. The CFC procedure took place over two consecutive days. On day 1, mice were placed in the conditioning chamber, and received 3 foot-shocks (1.5 s, 0.5 mA), which were administrated at 60, 120 and 180 s after the animals were placed in the chamber. They were returned to their home cages, 60 s after the final shock. Contextual fear memory was assessed 24 h after conditioning by returning the mice to the conditioning chamber and measuring freezing behavior during a 4 min retention test. Freezing was scored and analyzed automatically using Packwin 2.0 software (Bioseb, France). Freezing behavior was considered to occur if the animals froze for a period of at least two seconds.

### Brain harvesting and dissection

The day of birth was considered as P0. Animals were anesthetized by i.p. injection of 150 mg/kg of Ketamine (Imalgen 1000) and 20 mg/kg Xylazine (Rompun 2%), intracardially perfused with 4% paraformaldehyde (PFA) in 0.1 M PBS (pH 7.4) and post-fixed overnight in 4% PFA at 4 °C. Brains were cryoprotected overnight in 20% sucrose and embedded in O.C.T. compound (Tissue Tek, Sakura). Embedded brains were sectioned on a cryostat at 50 μm thickness. For immunodetection, sections were stored at −20 °C in EGG solution (33% Ethylene glycol, 33% Glycerol, 33% PBS). For in situ hybridization slices were dehydrated in an ascendant gradient of ethanol (25%, 50%, 75%, 100%) in PBS 0.1% Tween 20 and stored at −20 °C.

### Immunohistochemistry and immunofluorescence on floating sections

Brain slices were rehydrated in PBS and permeabilized in PBS-Triton 0.1% (PBT). For 3,3diaminobenzidine (DAB) staining, endogenous

peroxydases were inhibited with a solution of 3% $H_2O_2$ in PBT for 10 min. Unspecific sites were blocked in PBS, 0.2% gelatin; 0.25% Triton (PGT). Slices were incubated with the primary antibody: chicken anti-βgal (AbCam, 1:500) diluted in blocking solution, overnight (ON) at 4 °C. After rinsing in PBS, slices were incubated with a donkey biotinylated anti-chicken secondary antibody (Jackson, 1:500) for 45 min at RT. The signal was amplified with the Elite ABC (Vector) kit. To reveal the signal, a DAB solution prepared according to the manufacturer (Vector) was added to the slices, which were then mounted on glass slides in Mowiol. Immunofluorescent staining was performed as previously described[20,23,26]. As for primary antibodies: guinea pig anti-NPY (ab10341, Abcam, 1:500), rabbit anti-GABA (A2052, Sigma, 1:2000), rat anti-SST (MAB354, Millipore, 1:100), chicken anti-β-galactosidase (ab9361, Abcam, 1:4000), goat anti-Reelin (AF3820, R&D Systems, 1:500) and rabbit anti-p73 (ab40658, Abcam, 1:500) were used, while for secondary antibodies: donkey Cy3 anti-guinea pig (706-165-148, Jackson ImmunoResearch Laboratories, 1:700), donkey A488 anti-rabbit (711-545-152 Jackson ImmunoResearch Laboratories, 1:500), donkey Cy5 anti-rat (712-175-153 ImmunoResearch Laboratories, 1:250), donkey Cy5 anti-goat (705-175-147 ImmunoResearch Laboratories, 1:250), donkey A647 anti-rabbit (A32795, ThermoFisher, 1:200) and biotinylated goat anti-chicken (103-065-155 Jackson ImmunoResearch Laboratories, 1:1000) were used. DAPI (5 μg/ml, D1306, ThermoFisher Scientific) was used for fluorescent nuclear counterstaining of the tissue and mounting was done in Vectashield (H-1000, Vector Labs).

### In situ hybridization

Plasmid carrying DNA for the different probes were linearized using the appropriate restriction enzyme, purified using the Macherey–Nagel kit and cRNA probes (SST: 554 bp and GAD67: 982 bp) were synthesized using the appropriate RNA polymerase (Promega) and the DIG-labeling mix (Roche). The reaction mix was incubated for two hours at 37 °C and the probes were then purified using G50 columns (GE Healthcare) and stored at −20 °C in 50% formamide. Brain sections were rehydrated in a descending gradient of ethanol (100%, 75%, 50%, 25%) in PBS 0.1% Tween (PBT), washed twice in PBT, permeabilized with two 20 min washes in detergent mix solution (150 mM NaCl, 1% SDS, 1 mM EDTA, 50 mM Tris pH8.0, 0.5% Sodiumdeoxycholate, 1% NP-40) and postfixed in 4% PFA for 20 min at RT. After two washes in PBT, sliced were incubated at least 1 h at 70 °C with the pre-hybridization buffer (50% formamide, 5× SSC pH 4.5, 2% SDS, 2% BBR (Roche), 250 μg/ml yeast tRNA; 100 μg/ml heparin). Slices were then incubated ON at 70 °C with the hybridization mix. Sections were subsequently washed twice in 50% formamide, 2× SSC pH 4.5, 1% SDS at 70 °C for 30 min, then 3 times in MABT solution (100 mM maleic acid, 150 mM NaCl, 0.1% Tween) at RT, followed by a blocking step of at least 1 h in 20% horse serum in MABT at RT. Alcaline phosphatase conjugated anti-Digoxigenin antibody (Roche) diluted 1:2000 in blocking solution was added overnight at 4 °C. Slices were washed 3 times in MABT solution at RT for 30 min, and then they were incubated with NTMT solution (100 mM NaCl, 100 mM Tris-HCl pH9.5, 50 mM $MgCl_2$, 0.1% Tween-20) for 30 min. Staining solution (450 ng/ml of Nitro-blue tetrazolium (NBT) and 175 ng/ml of 5-bromo-4-chloro-3'-indolyphosphate p-toluidine salt (BCIP) in NTMT) was added to the slides and the color reaction was allowed to develop in the dark at RT. The reaction was stopped and slices mounted on slides using Mowiol.

### Image acquisition and cell counting

Immunofluorescence images were acquired using a confocal microscope LEICA TSC SP8 inverted confocal microscope with PL APO ×20/0.75 NA ×20 or PL APO ×40/0.85 CORR CS,0.11- oil immersed objectives (without digital zoom). Confocal images consist of multiple tile regions (mosaics) combined with serial z-stacks (0.5 μm through all the section's depth). Composite images are presented as maximum

projections and were generated by the LAS X software using Mosaic Merge and Projection functions. Discontinuities are due to mosaic stitching algorithm. Images were acquired with a 12 bit resolution of 1024 × 1024 pixels using bidirectional laser line average of 1 and combined with z-stack series at intervals of 3 μm. IHC and ISH images were acquired using a slide scanner Nanozoomer 2.0 (Hamamatsu) with a ×20 objective. Number of CRs or different subtypes of interneurons were counted using the ImageJ software in the different hippocampal regions (CA1/CA2/CA3/dDG (HF) and vDG for CRs and CA1/CA2, CA3 and DG for interneurons, identified based on histology) for each age and genotype, at bregma −1.82. The left and right hippocampus were counted and the mean value was used as n = 1. For CRs, all the cells along the HF (SLM for CA1/CA2/CA3 and OML for dDG) and vDG (OML) were counted. The total number of cells was divided by the length of the HF or vDG to obtain the cells/mm value. For interneurons, the total number of cells in the different regions (CA1/CA2, CA3 and DG) was counted and divided by the total area of that region to obtain the cells/mm² value.

### Western blot analysis

Brains from 3 months old adult mice were dissected, after euthanizing the animals by cervical dislocation, and collected in ice cold PBS. Hippocampus cell lysates from both hemispheres were collected using 50 μl RIPA buffer (50 mM Tris-HCl pH 7.5, 150 mM NaCl, 1% NP-40, 0.1% SDS, 0.5% sodium deoxycholate) supplemented with 2 mM EDTA and protease inhibitors (cOmplete™ EDTA-free tablets). Samples were then left in a rotator for 30 min at 4 °C to better dissolve the proteins and centrifuged at max speed for 10 min at 4 °C in order to pellet out crude undissolved fractions. The supernatant was stored at −20 °C until use. Protein quantification was determined using the bicinchoninic acid (BCA) protein assay reagent kit (Pierce™, ThermoFisher Scientific, USA) using bovine serum albumin (BSA) as standard. Proteins samples were added to 1/4 volume of 4X LDS sample buffer (141 mM Tris, 106 mM Tris-HCl, 2% LDS, 10% glycerol, 0.51 mM EDTA, 0.22 mM SERVA Blue G, 0.175 mM phenol red, pH 8.5), to 1/10 of sample reducing agent 10X (50 mM DTT) and boiled at 95 °C for 3 min. Protein samples (40 μg) were separated by SDS-PAGE on 3–8% tris-acetate gels (NuPAGE, Invitrogen) under reducing conditions at 150 V at room temperature (RT) and electro-transferred to 0.45 μm nitrocellulose membranes (Amersham, GE Healthcare, Germany) for 1 h 30 min–2 h at 0.5 A at 4 °C. After 1 h blocking at RT with 5% (w/v) milk in Tris-buffered saline (50 mM Tris, 150 mM NaCl, pH 7.6) containing 0.1% Tween 20 (TBS-T), membranes were incubated overnight at 4 °C with the following primary antibodies: mouse anti-RELN G10 (MAB5364, Millipore 1:2000) and rabbit anti-alpha tubulin (PA529444, Invitrogen 1:1000) as loading control. After three washing periods of 10 min with TBS-T, the membranes were incubated with the appropriate HRP-conjugated secondary antibodies (Jackson Immunoresearch; 1:20,000) diluted in TBS-T with 5% (w/v) milk for 1 h at RT. After three 10 min washes with TBS-T, blots were developed with SuperSignal West Pico Chemiluminescent Substrate and visualized on the Chemi-Doc apparatus (Bio-Rad). Quantitative analysis was determined by densitometry using Image Lab™ v3.0 software (Bio-Rad). The densities of protein bands (including all Reln cleavage products) were quantified with background subtraction. The bands were normalized to alpha-tubulin loading control. The molecular weights were determined by using an appropriate pre-stained protein standard (HiMark 31–460 kDa, Invitrogen) for high molecular weight proteins.

### In vivo electrophysiology: awake head-fixed preparation

All experiments were performed on adult mice (BaxCKO mutants and control littermates). Briefly, as previously reported in Navas-Olive et al.[41], mice were implanted with head fixation bars (anterior to bregma) and ground/reference pins (over the cerebellum) under anesthesia (isofluorane 4% for induction and 1–2% for maintenance).

Both bars and reference pins were fixed with dental cement. After surgery, mice were returned to their group-housed home cages and habituated to the head-fixed wheel apparatus over a period of 2–3 weeks. The apparatus consisted of a spherical foam wheel/treadmill (40 cm diameter) equipped with a sensor to estimate speed and distance traveled analogically on which animals were trained and habituated for 2 sessions per day until fully habituated. Animals were deemed habituated when they ran freely and consistently at intervals throughout training sessions and comfortably stayed in the head-fixed preparations for up to 2 h with intermittent periods of running, grooming and rest.

Once habituated to the apparatus, animals were anesthetized with isoflurane as above to perform a craniotomy for electrophysiological recordings (antero-posterior (AP): −1.8 to −2.3 mm from Bregma; medio-lateral (ML): 1.2–1.7 mm). The craniotomy was sealed with Kwik-Cast silicone elastomer and animals returned to their home cage. In vivo electrophysiological experiments started the day after craniotomy.

For LFP recordings in head-fixed conditions, we used 16-channel silicon probes from Neuronexus (linear arrays with 100 μm resolution and 413 μm² electrode area). LFP signals recorded at individual channels were pre-amplified (4–10 × gain), recorded with multichannel amplifiers (Multichannel Systems) and digitized using a Digidata 1440A, Molecular Devices. Characteristic features such as the laminar profile of theta and sharp-wave ripples were used to inform electrode position within the hippocampus. Electrode position was later confirmed histologically.

### Spectral analysis of LFP activity

All analysis was performed using routines written in Matlab 2021a (MathWorks). For experimental data, LFPs from different layers were identified according to distinctive features, including sharp-wave ripples (at SR and SP) and maximal theta oscillations (at SLM). Power spectra of LFP signals were estimated using the Fast Fourier transform (FFT) for theta (4–12 Hz) and gamma (40–60 Hz) over appended periods of mobility. In order to detect mobility intervals, power spectrum was computed for windows of 5 s every 0.5 s in the channel of maximum theta power (usually SLM channel). For every window, mean power for theta (4–12 Hz) and delta (0.1–4 Hz) was obtained, and a theta over delta ratio was computed. High values of this power ratio are related to theta states, associated with mobility periods, while low values are related to large irregular activity (LIA) states, more frequent during resting or grooming. Theta over delta ratios usually result in a bimodal distribution, so we consider mobility intervals those in which the theta over delta ratio is over a threshold adjusted to the bimodal distribution. Contributions of 50 Hz and harmonics were filtered out and data between the filter limits were interpolated for visualization purposes only.

### Electrode track verification and CR quantification

Mice were perfused with 4% paraformaldehyde and 15% saturated picric acid in 0.1 M, pH 7.4 phosphate buffered saline (PBS). Brains were post-fixed overnight and serially cut in 50 μm coronal sections (Leica VT 1000S vibratome).

Each electrode track was identified by dark-field microscopy (Leica S8APO). To evaluate the density of persistent CRs around electrode tracks, selected sections were stained against CR-specific marker P73 (see details above in Immunohistochemistry) and mounted on glass slides in ProLong Gold (Invitrogen) for subsequent fluorescence imaging (LEICA AF 6500/7000). A N2.1 filter (excitation, dicroic, emission spectral filters: BP515-560, LP590, 580) and a 10 × 0.3 dry objective were used. The Fiji software (NIH Image) was used for image adjustment and analysis. Quantification was performed blindly with respect to electrophysiological data. P73⁺ cells around the hippocampal fissure comprised within CA1 were counted in sections

containing electrode tracks. P73$^+$ cells around the HF were counted on both sides of the HF: in CA1 SLM as well as in the DG ML. The total number of P73$^+$ cells around the CA1 HF was divided by the total length of the CA1 HF to obtain a cell density value (cells/mm) per section.

Some sections were washed in 1% Triton X-100 (Sigma) in PBS and incubated 20 min with bisbenzimide H33258 (1:10,000 in PBS, Sigma, B2883) for nuclei labeling. Multichannel fluorescence stacks were obtained in a confocal microscope (Leica SP5) with the LAS AF software v2.6.0 build 7266 (Leica), a HC PL APO CS 10.0×0.40 DRY UV objective, and the following channels (fluorophore, laser and excitation wavelength, emission spectral filter): a) bisbenzimide, Diode 405 nm, 415–485 nm; b) TdTomato, DPSS 561 nm, 571–620 nm.

### In vitro electrophysiology: patch-clamp recordings and analysis

**Patch-clamp recordings and analysis.** Acute coronal slices (320 μm) were prepared from adult *BaxCKO* control or mutant mice, starting from postnatal days 100 to 120. Mice were deeply anesthetized with a mix of ketamine/xylazine and perfused intracardially with ice-cold slicing solution with the following composition (in mM): 220 sucrose, 11 glucose, 2.5 KCl, 1.25 NaH$_2$PO$_4$, 25 NaHCO$_3$, 7 MgSO$_4$, 0.5 CaCl$_2$. After transcardiac perfusion the brain was quickly removed and coronal slices (320 μm) were prepared using a vibratome (Leica VT1200S). After slicing, brain slices were transferred to ACSF solution maintained at 34 °C containing (in mM): 125 NaCl, 2.5 KCl, 2 CaCl$_2$, 1 MgCl$_2$, 1.25 NaH$_2$PO$_4$, 25 NaHCO$_3$, 15 glucose. Following a 15–20 min incubation period slices were kept at room temperature for a maximum of 6 h. Whole-cell patch-clamp recordings were performed using a Multiclamp 700B amplifier (Molecular Devices) and fire-polished thick-walled glass patch electrodes (1.85 mm OD, 0.84 mm ID, (World Precision Instruments, ref.1B150F-4); 3.5–5 MOhm tip resistance) close to physiological temperatures (33–35 °C). For voltage clamp experiments cells were whole-cell patched using the following intracellular solution (in mM): 110 Cs-MeSO$_3$, 1 EGTA, 10 HEPES, 4 MgCl$_2$, 3 QX-314,10 Na$_2$Phosphocreatine, 0.3 MgGTP, 4 Na$_2$ATP (300 mOsm pH adjusted to 7.3 using CsOH). Biocytin (10%) was added to intracellular solution to allow post hoc morphological reconstruction of recorded neurons. sEPSCs and sIPSCs were recorded holding CA1 pyramidal cells at −60 mV and 0 mV respectively. All recordings were low-pass filtered at 10 kHz and digitized at 100 kHz using an analog-to-digital converter (model NI USB 6259, National Instruments, Austin, TX, USA) and acquired using Neuromatic v3 (Rothman and Silver, 2018) running in Igor PRO 9 (Wavemetrics, Lake Oswego, OR, USA). Data analysis was performed using the Neuromatic analysis package (Rothman and Silver, 2018) written within the Igor Pro 9 environment (WaveMetrics, Lake Oswego, OR, USA). Recordings were offline filtered at 3 kHz and sEPSCs, mEPSCs and sIPSCs were detected using a threshold detection method (amplitude higher than two times the standard deviation of the filtered baseline). Average frequency of sEPSC, mEPSCs or sIPSCs was calculated for individual CA1 neurons using all the events detected within a 4 min recording period. For all the experiments, amplitude of synaptic currents was calculated as the average amplitude of a total of 100 events. We excluded from the analysis events in which a second peak occurred in the rising phase of the first detected synaptic current. Recordings were not corrected for liquid junction potential. For all experiments data were discarded if series resistance, measured with a −5 mV pulse in voltage clamp configuration was higher than 20 MOhm or changed by more than 20% across the course of the experiment. Miniature excitatory post-synaptic currents (mEPSCs) were recorded at a membrane potential of −60 mV in the presence of TTX 1 μM. For current clamp recordings, a potassium-based intracellular solution was used (in mM): 135 K-D-gluconate, 5 KCl, 10 HEPES, 0.01 EGTA, 10 Na2 phosphocreatine, 4 MgATP, 0.3 NaGTP (pH = 7.4, 295 mOsm). Intrinsic properties of CA1 neurons were measured in current-clamp configuration using 1-s-long current injection step. The resting membrane potential (RMP, in mV) was measured with 0 pA current injection 3 min after obtaining whole-cell access. Currents steps were injected at RMP. Input resistance (Ri) was estimated from a hyperpolarizing step upon the injection of −10 pA to cells at RMP.

### Analysis of CA1 pyramidal neuron morphology

Slices containing biocytin filled single pyramidal cells were fixed in 4% PFA for at least one hour. Slices were washed in PBS 3 times 10' at RT and then 3 times in blocking solution (4% NGS (normal goat serum), 0.2% Triton in PBS) at RT. Slices were then incubated with Streptavidin-Alexa488 diluted 1:500 in 3%NGS, 0.2% Triton in PBS for 2 h at room temperature (RT). Three rinses in PBS 3 times 20' at RT were performed before mounting in Vectashield. Pyramidal neurons were acquired using a LEICA SP8 confocal microscope with a ×40 objective and a 1.3 digital zoom. For dendritic spines, images of apical and basal dendrites were acquired with a ×63 objective, 2.5 digital zoom using fixed parameter of pinhole aperture, gain and laser intensity. For apical dendrites, a terminal ramification in the SLM was acquired, while for basal dendrites a descending dendrite was acquired approximately at the same distance from the soma. 3D reconstruction was performed using the IMARIS software 8.4. Dendrites exiting from the top of the soma were considered apical, while all the others were considered basal. Statistical analyses were performed based on the data given by IMARIS in apical and basal dendrites separately. Quantification of spines was performed manually using ImageJ v1.53k software on black and white maximum projections.

### Video-EEG recordings

6 females and 9 males were used for this experiment. Experiments were performed when the animals were aged 4-5 weeks and weighted 18–22 g on the date of surgery. They were housed in individual cages with food and water *ad libitum*, and kept in a 12-h light/dark cycle after surgery. All experiments were performed in accordance with the rules of the European Committee Council Directive 2010/63/EU, after validation by our local ethical committee and authorization from the French Ministry of Research. All efforts were made to minimize animal suffering and reduce the number of animals used in each series of experiments.

Mice were stereotaxically implanted under general anesthesia (chloral 4%) with a bipolar electrode made of polyester-insulated stainless steel (diameter, 0.125 mm, FE245840, Goodfellow), placed in the right dorsal hippocampus [anteroposterior (AP), −2; mediolateral (ML), −1.5; dorsoventral (DV), −2] with bregma as the reference (Paxinos and Franklin, 2001), as previously described[57,82,83]. A monopolar electrode was positioned over the left fronto-parietal cortex and a monopolar electrode inserted above the cerebellum (reference electrode)[57,82,83]. The electrode/connector assembly was maintained on mouse heads with cyanoacrylate glue and dental acrylic cement. Video-EEG activity was recorded and analyzed with a digital acquisition computer-based system (Coherence, Deltamed, France; sampling rate 1,024 Hz) during several recordings sessions performed in freely moving animals between 1:00 and 5:00 p.m. Mice were first connected to the recording system and habituated to their test cage for 1 h, then hippocampal and cortical EEG activities were recorded for 3 h (3 sessions), one 6-h session during the day shift and one 12-h session during the night shift over the 2 weeks that followed surgery. Digitally recorded EEGs were analyzed, using a referential setup with the electrode placed over the cerebellum. All analyses were performed by experimenters blind to animals' genotype.

### Kainate injection and recording

Animals were injected with a single i.p. dose of Kainate (#0222/10, Biotechne) at 15 mg/kg and were recorded immediately via a CaptureStar v1 software for 2 h. The latency of the first seizure, the total

duration of all seizures, and the number of seizures were recorded by a user blinds to the animal genotype. The mortality due to the seizure was also noted.

## Multi-electrode array (MEA) recordings

All experiments were performed on adult mice. Mice were sacrificed and the brain removed. Horizontal cortico-hippocampal slices (400 μm) were cut at low speed (0.04 mm/s) and at a vibration frequency of 70 Hz in ice cold oxygenated artificial cerebrospinal fluid (ACSF) supplemented with sucrose (in mM: 87 NaCl, 2.5 KCl, 2.5 CaCl$_2$, 7 MgCl, 1 NaH$_2$PO$_4$, 25 NaHCO$_3$ and 10 glucose, saturated with 95% O$_2$ and 5% CO$_2$). Slices were maintained at 32 °C in a storage chamber containing standard ACSF (in mM: 119 NaCl, 2.5 KCl, 2.5 CaCl$_2$, 1.3 MgSO$_4$, 1 NaH$_2$PO$_4$, 26.2 NaHCO$_3$ and 11 glucose, saturated with 95% O$_2$ and 5% CO$_2$) for 20 min, and then stored for at least one hour before recording in a pro-epileptic (0 mM Mg$^{2+}$, 6 mM K$^+$) ACSF.

Cortico-hippocampal slices were transferred on planar MEA petri dishes (200-30 ITO electrodes organized in a 12 × 12 matrix, with internal reference, 30 μm diameter and 200 μm inter-electrode distance; Multichannel Systems, Germany), and kept in place using a small platinum anchor. The slices were continuously perfused at a rate of 2 ml/min with 0Mg6K ACSF during recordings. Pictures of cortico-hippocampal slices on MEAs were acquired with a video microscope table (MEA-VMT1; Multichannel Systems, Germany) through MEA Monitor v1.0.5 software (Multichannel Systems, Germany) to identify the location of the electrodes on the cortex and hippocampus and to select electrodes of interest. Data were sampled at 10 kHz and network activity was recorded at 32 °C by MEA2100-120 system (bandwidth 1–3000 Hz, gain 5x, Multichannel Systems, Germany) through the MC Rack 4.5.1 software (Multichannel Systems, Germany).

## Multi-electrode (MEA) data analysis

Bursting raw data were analyzed with MC Rack (Multichannel Systems, Germany). Detection of bursts was performed using the "Spike Sorter" algorithm, which sets a threshold based on multiples of standard deviation of the noise (5-fold) calculated over the first 500 ms of recording free of electrical activity. A fivefold standard deviation threshold was used to automatically detect each event, which could be modified in real-time by the operator on visual check if needed. Bursts were arbitrarily defined as discharges shorter than 5 s in duration. Typically, bursts were characterized by fast voltage oscillations followed by slow oscillations or negative shifts. To analyze seizure activity, data were exported to Neuroexplorer v4 (Nex Technologies, USA). Paroxysmal events were identified as discharges lasting more than 5 s; successive paroxysmal discharges were considered separate events based on their waveform and on the presence of a minimum (>10 s) interval of silent or bursting activity between them[84].

## Statistical analysis

Statistical parameters including the exact sample size (n), post hoc tests, and statistical significance are reported in every figure and figure legend. Data were estimated to be statistically significant when $p \le 0.05$ by Student's $t$ test, Mann–Whitney test, Chi-square test Wilcoxon sign test or two-way ANOVA. Statistical analyses were performed using GraphPad PRISM (v5) (GraphPad Software Inc., La Jolla, CA, USA). All values are shown as mean ± SEM. Source data are provided as a Source data file.

## Reporting summary

Further information on research design is available in the Nature Portfolio Reporting Summary linked to this article.

## Data availability

All data are available in the main text or Supplementary Materials. Source data are provided with this paper.

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

## Acknowledgements

We thank Dr. P. Billuart for critical reading of the manuscript and suggestions during the course of the study, the NeuroImag platform at the IPNP and SFR Necker Imaging and histology platforms at the *Imagine* Institute for help with acquisition, the animal house facility (LEAT) and Animalliance for animal care. We are grateful to N. Ramezanidoraki and P. Billuart for initiating the first MEA experiment as well as members of the Pierani's lab for technical support and helpful discussions. We thank Ann Kennedy for mouse profile (Zenodo, 2020) doi:10.5281/zenodo.3925921and for the mouse scheme in Fig. 3a, French Ministry of Research (BioSPc Doctoral school) (M.R.), Fondation pour la recherche médicale, FDT20201201037 (M.R.), Centre national de la recherche scientifique (CNRS) (A.P.), Agence Nationale de la Recherche, ANR-15-CE16-0003-01, ANR-19-CE16-0017-03 and ANR-20-CE16-0001-01 (A.P.), Fondation pour la recherche médicale, Équipe FRM DEQ20130326521 and EQU201903007836) (A.P.), Agence Nationale de la Recherche under "Investissements d'avenir" program, ANR-10-IAHU-01) (*Imagine* Institute), Fondation pour la recherche médicale (F.O.), AGEMED-INSERM (F.O.), NRJ for Neuroscience (F.O.), European Research Council (Consolidator grant #683154) (N. Rouach), European Research Council (Starting Grant #678250) (N. Rebola), Agence Nationale de la Recherche ANR-21-CE16-0020 and ANR-20-CE16-0009 (N. Rebola), and ANR-21-NEU2-0007-01 Eranet-Neuron ROSSINI project (A.P.) and PCI2021-122080-2A by MCIN/AEI/10.13039/501100011033 and European Union "NextGenerationEU"/PRTR" (L.M.d.l.P.).

## Author contributions

Conceptualization: M.R., S.M., F.O., and A.P. Methodology: M.R., S.M., A.M., E.D., F.D., F.L., S.D., F.T., M.B., N. Rebola, A.D., E. Coppola, N. Rouach, L.M.d.l.P., F.O., and A.P. Investigation: M.R., S.M., A.M., E.D., C.S.-B., P.A., A.N.-O., B.G., N. Rebola, E. Cid, A.D., E. Coppola, N. Rouach, L.M.d.l.P., F.O., and A.P. Data curation: M.R., S.M., A.M., E.D., C.S.-B., P.A., A.N.-O., N. Rebola, A.D., E. Cid, N. Rouach, L.M.d.l.P., F.O., and A.P. Writing—original draft preparation: M.R., S.M., L.M.d.l.P., F.O., and A.P. Writing—review and editing: M.R., S.M., A.M., E.D., C.S.-B., P.A., N. Rebola, A.D., E. Coppola, N. Rouach, L.M.d.l.P., F.O., and A.P. Visualization: M.R., S.M., E.D., C.S.-B., N. Rouach, F.O., and A.P. Supervision: A.P. Project administration: A.P. Funding acquisition: A.P., L.M.d.l.P., N. Rebola, and F.O.

## Competing interests

The authors declare no competing interests.
