## [Peer Review File · Nature Communications]

Aberrant survival of hippocampal Cajal-Retzius cells leads to memory deficits, gamma rhythmopathies and susceptibility to seizures in adult miceREVIEWER COMMENTS

Reviewer #1 (Remarks to the Author):

The manuscript by Riva and colleagues investigates the consequences of the aberrant survival of Cajal-Retzius cells (CRs) on hippocampal-dependent behaviors, network oscillations, and susceptibility to seizures. The authors take advantage of a previously established animal model with the invalidation of the pro-apoptotic factor Bax, which allows the unphysiological survival of CRs.

Results show that the aberrant survival of hippocampal CRs is associated with network alterations at the structural and functional level, cognitive deficits, and an adult epilepsy-prone phenotype.

In general, the work is very nice and addresses an important question with high translational impact because the persistence of CRs has been reported in different pathologies. Furthermore, CRs are still a relatively mysterious neuronal type, and therefore this work has the potential to significantly advance the field. There are, nevertheless, points that require improvements.

1) The discussion is too descriptive, does not address potential mechanistic scenarios, and does not clarify apparent inconsistencies in the results. In more detail:

a) Are the effects observed in mutant mice caused by increased reelin availability due to CR survival or by computational effects driven by increased glutamatergic output due to rescued CRs? The increased complexity of pyramidal cell dendritic trees would suggest a contribution of reelin-mediated signaling. Is anything known about reelin levels in the hippocampus of the BaxCKO mouse? Could these be measured and compared to control animals? Regarding computational effects, have the authors attempted optogenetic stimulation of rescued CRs to evaluate changes of EPSCs in interneurons (as previously reported in the neocortex, ref 29)? These points should be at least discussed.

b) Some of the behavioral changes observed in mutant mice are very similar to what was recently reported in mice with inactivated glutamatergic functions of CRs (ref 28). This suggests that either loss or gain of function of hippocampal CRs are sufficient to impair circuits underlying memory processing. Could this consideration please be included in the discussion?

c) Spine density increase in pyramidal cells with no parallel changes in spontaneous or miniature EPSC frequency. How do the authors interpret this apparent inconsistency? Do they think that the extra spines do not receive synapses and/or remain at an immature stage? Also, this seems to contrast with what was found in the neocortex of the same mutant mice, where a parallel increase in spine density and sEPSC frequency was observed (ref 29). This point should be explicitly mentioned and discussed.

d) Analysis of network oscillatory patterns. Firstly, the decrease in theta power in mutant mice appears to be selective for SLM recording sites, in contrast to gamma power, which was found to increase both in SLM and SP. What is the explanation for this selectivity? Secondly, it is unclear how the correlations (or lack thereof) between theta/gamma power and CR counts should be interpreted (Fig 3E). The data show a significant correlation only for theta power recorded in SP (but not in SLM). The lack of

correlation between SLM theta and CR counts is counterintuitive and difficult to understand, given the observed differences between the power of SLM theta in control vs. mutant mice (which is presumably due to the different numbers of CRs). Lastly, given the significant correlation between SP theta power and CR counts wouldn't one expect to see a difference in SP theta power between control and BaxCKO mice (larger population of CRs)? Please clarify.

e) The methods section indicates that patch-clamp recordings were obtained from adult animals (P100-P120) and Fig 4 shows that enhanced survival of interneurons in mutant mice is specific for juvenile, but not for adult animals. How can a transient increase in interneuron survival (at juvenile stages) explain the increased frequency of sIPSCs and theta-gamma rhythmopathies found in adult mutant animals? As the interneuronal population in adult mutant animals (apparently) does not differ from control mice, the most logical explanation would seem to be an increase in the excitatory drive from (an augmented population of) CRs onto interneurons. This point should be explicitly discussed.

f) The only significant functional alteration reported in hippocampal pyramidal neurons of mutant mice seems to be an increase in spontaneous inhibition, whereas both intrinsic excitability and spontaneous excitation remain unchanged. This would suggest an overall decrease in the E/I balance, which is usually associated with a reduced probability of generating seizures and would seem in contrast to the observed lowered seizure threshold. How do the authors explain this apparent inconsistency? Were evoked IPSCs and EPSC affected in mutant mice? This point should be ideally tested experimentally or at least discussed.

g) The significance of the experiments in Figure 7 is rather unclear. Are the authors suggesting that seizures originate in the entorhinal cortex and, from there, they then propagate to the hippocampus? A comparison of the activity in the different channels should be able to identify the site(s) of initiation of the seizure-like activity. Could this point please be addressed, and results discussed?

2) Figure 1A. The upper left and right panels are identical. Please correct. Also, it is unclear why data for control animals are reported exclusively as a plunger plot, whereas individual results are also displayed for mutant animals. For consistency, please show individual results for both groups. In the text, it is stated that this experiment was performed in 2 assays at 2 min distances, but it is unclear what is actually represented in the plot (1st assay only, 2nd assay only, average of the two assays). Please clarify.

3) Figure 1A. Bottom right panel. The latency to find food tested at a 24-hour interval is reported as not significant for male mutant mice. However, these data are very similar to what was found for male control animals. Are the authors sure of this result? In particular, it is unclear what statistical test was used, as in the text it is mentioned that a two-way ANOVA with post hoc Sidak's multiple comparison test was used, whereas in the legend it is stated that a Mann-Whitney test was used. As this was a paired comparison (1st vs 2nd trial in the same animals), shouldn't a Wilcoxon signed-rank test be used? Would this test result in significance?

4) Figures 2B (and 4D). It is very hard to see CRs (interneurons) on printouts. Could the authors please either increase the contrast or include insets at increased magnification?

5) Figure 3. The vertical size of the figure is rather small, which makes it difficult to see the data. Also, the font size of the axis labels is too small to read. Please increase the size.

6) Figure 6. Please include examples of EEG recordings.

7) In general, the consistency of the illustrations should be improved by using the same type of plot and convention. For example, could the authors please use either plunger plots plus individual results (as in Figs 1B, 2, 4, etc..) or mean and error bars plus individual results (as in Figs 1C, 1F, 5C, 5D, etc...) or boxplots and individual results (Fig 3) throughout the paper? Also, could the same convention to distinguish control (white bars and empty circles) from mutant mice (blue bars and black circles) please be used for every figure?

8) Page 3. "These cell populations are unique as they completely disappear at the end of cortical maturation shortly after birth in mice". This statement (referring to CRs in general) is incorrect as hippocampal CRs persist in adult animals and never completely disappear. Please reword this sentence.

9) Page 3. "Strikingly, transient hippocampal CRs stay longer in juvenile mice and yet a small 15% remain integrated in adult microcircuit". This sentence is inherently contradictory as it claims that "transient" neurons "remain in adult microcircuits". Please correct. Also, I am not sure how the 15% was calculated as none of the references quoted (refs 27, 28) address quantitatively this specific point. Reference 37 shows that from P7 to adulthood a ~30% of hippocampal CRs persist. Please modify.

10) Page 4. "A primary screen of 65 tests [...] was performed". However, only a much more limited number of tests is presented in Supplementary Table 1. If 65 different tests were performed, then these results should be included in a Table.

11) Data presentation. Could the actual data values please be included either in the text or in a supplementary table? Most of the time, results are presented in the text only by indicating the statistical test that was performed and its probability. Also, it is often quite difficult to find the exact sample size (n) and unit of measurement for each experiment. Could the authors please consistently include these with the numerical measurements, test performed, and probability (either in the text or in a supplementary table)?

Reviewer #2 (Remarks to the Author):

In this interesting and very thorough manuscript, the authors utilize a genetic mouse model of Bax in activation in p73 expressing cells to suppress programmed cell death of Cajal Retzius (CR) cells derived from the septum and from cortical hem. The authors use this model to assess the effects of CR

persistence on behavior, and cellular and circuit activity in the hippocampus. They also report impairment in hippocampus-dependent behaviors, specifically spatial and episodic-like memory in adults, but not juveniles. These behavioral changes were associated with increases in theta oscillations, but decreases in gamma activity. There was also a decrease in NPY-GABAergic cells in juveniles and a changes in pyramidal cell spine density in adults. Mutant mice also had increased likelihood of tonic-clonic seizure activity.

In general, this was an excellent manuscript. There is an exhaustive list of behavioral tasks performed and the results range from behavior to circuit activity to neural pathology. The results contribute enormously to our understanding of the role of transient CR cells in hippocampal development and function and demonstrate that their demise via programmed cell death is as important to normal function as their transient role in establishing circuit connectivity. However, there are a few concerns with the manuscript that should be easily addressed.

Can the behavioral tasks performed be disentangled from cortical function? Presumably, the septum CRs in cortex are spared in this mouse model? For example, novelty detection relies on hippocampal/mPFC circuits whereas object recognition relies heavily on lateral entorhinal cortex (LEC) and upstream perirhinal connectivity with PFC. Deficits on this task suggest impaired LEC processing. Perhaps CR survival in regions “upstream” of the hippocampus is contributing to the observed effects as well? The addition of some language to address this possibility, or at least toning down the claims that the behavioral deficits are definitely and solely due to CR persistence in hippocampus is warranted.

There was no effect on contextual fear conditioning which is hippocampus dependent including CA1. Can the authors address why they think this behavior is normal despite their other neural and behavioral findings?

In Figure 1A, bottom panel: the statistical analyses are performed separately for controls and mutant mice, demonstrating that there is a significant decrease in latency between Trials 1 and 2 in controls, but not in mutant mice. However, the Y axis are different on the control and mutant mice graphs. It appears that mutant mice are faster on Trial 1 than controls, but do not really differ much in the latencies on Trial 2, suggesting that there is no significant decrease in latency in mutant mice because they are already performing quickly in Trial 1. This leads to a different conclusion than the authors make. The suggestion is to conduct a two-way (Trial X mutation) ANOVA to assess this issue.

On page 6, the authors report that they were unable to discriminate between the SLM of the CA1 and the outer molecular layer of the dDG. They go on to say that because no differences were found in the vDG, the surviving CRs must be in CA1 SLM. This is not a sound conclusion as dorsal and ventral DG (i.e., infra- and supra blades of the outer molecular layer) can have different functions. Also, the explanation about combining the CA1/CA2/dDG should be included in the caption of Figure 2. As is, it appears that CRs are being counted in CA1 and CA2 proper (not in SLM), but clearly there are no β gal cells in the pyramidal cell layer of those regions.

On page 10, “CRs persistence leads to enhances susceptibility to seizures”: This paragraph refers to data from analysis of video-EEG recordings. Are these data shown in the manuscript?

In Figure 6A and 6C: The latency to seizure did not differ between control and mutant responders. However, Figure 6A shows a significant difference in latencies (180 vs 24). Does this mean that non-responders were assigned the maximum latency to get the results shown in 6A? This seems to already be covered in the seizure incidence data and creates some confusion. It should be explained more clearly in the figure caption and/or results or latency in non-responders should be removed.

On page 12, reference 49: Based on the title of the reference this study examined a progesterone receptor antagonist and not estrogen treatment. The authors should check this.

In general, the manuscript should be checked for grammar and typos.

Reviewer #3 (Remarks to the Author):

The present manuscript entitled “Aberrant survival of hippocampal Cajal-Retzius (CR) cells leads to memory deficits, gamma rhythmopathies and susceptibility to seizures in adulthood” by Riva et al. builds on previous work of the same group investigating the persistence of Cajal Retzius cells (CR) in the neocortex. CR normally undergo programmed cell death (PCD) after completion of neocortical networks. The results of this study were obtained in mouse mutants (BaxCKO Δ Np73Cre = BaxCKO) in which Bax-dependent PCD of a CR subtype is prevented. A battery of behavioral tests was performed with these mutants (males and females) and compared to wild type animals. Authors found in particular that hippocampus-dependent cognitive functions were normal in young adult, but affected in adult mutant mice of both sexes as shown by a Cookie test, Morris water maze, novel object recognition and object location tests. Authors concluded that the persistence of transient CR may induce a major impairment of hippocampal function. Therefore, they investigated the pattern of CR cells in their mutant mouse line and found an increased survival of LacZ-expressing cells (= CR cells) in the adult hippocampus. Based on these observations they conducted in vivo LFP recordings with 16-channel silicon probes and in vitro patch-clamp and a multi-electrode array (MEA) recording to determine potential consequences of CR cell survival. They found an attenuation of theta oscillations and enhanced gamma activity in the dorsal CA1 region, transient changes in the number of NPY+ GABAergic cells in juvenile mice and alterations of CA1 pyramidal cells spine density in adults, enhanced frequency of inhibitory currents in adult hippocampal pyramidal cells and an increased susceptibility to kainate-induced status epilepticus.

Results of this study are basically interesting. Several state of the art techniques (behavioral tests, immunocytochemistry, in situ hybridization, patch-clamp recordings, multi-electrode array (MEA) recording and in vivo LFP recordings with 16-channel silicon probes) have been applied but this referee has concerns regarding the nature of the surviving hippocampal neurons in BaxCKO mouse mutants. The aberrant survival of CR cells into adulthood is the basis for all follow-up experiments in this manuscript

and should be more convincingly demonstrated. See specific comments and concerns below.

Introduction:

Persistence of CR cells in TLE with AHS as stated in the introduction is controversial, there is only one study reporting an increase, whereas many papers demonstrate a loss of CR cells in human patients as well as in rodent epilepsy models.

Result and Legends:

In the manuscript, three different age groups (P24, 9 weeks and 3 months) have been analyzed. This should be indicated clearly and consistently in the Figures and Legends. It is quite tedious and confusing to get it in the present manuscript. In addition, there is quite a bit of redundancy in Legends, they could be shortened substantially.

Behavioral tests:

Morris Water Maze (MWM) test: is this really appropriate for mice, since they are afraid of water and consecutively very stressed? A dry land maze such as the Barnes mazes would be more suitable.

Fig. 1: Were the same mice tested in the different behavioral paradigms? Numbers of mice are different in the different tests.

Why only the males are shown in Fig. 1? Data on females are all put into the supplement, although they show also deficits in memory.

Concerning female mutants: only tests performed with 3-month-old mice are shown. Is the age-dependent switch from normal behavior in young adults to impaired memory in older ones also present in females? As far as I understood the plethora of behavioral tests, females were only tested at young age and in the adult stage.

Fig. 2. What is juvenile and what is adult? In B it is stated P24 hippocampus, is this correct?

My main concern with this Figure is that the quality of the micrographs in B is poor. Labeled cells cannot be recognized. In addition, why should there be CR cells in CA3? CR cells are usually confined to the hippocampal fissure. It is also surprising that the number of LacZ-positive cells is increased in CA3 in the adult stage when compared to juvenile animals. How can this be explained?

Did the authors verify the CR-specificity of the LacZ-expression with CR-markers? Calretinin or reelin could be used to label CR cells at the hippocampal fissure (see Anstötz et al., 2018 and others).

Fig. S3. This referee does not understand the rationale of using BaxCKOwnt3acre mice. They express LacZ not only in CR cells but also in granule cells and pyramidal cells as obvious in Fig. S3.

Fig. 4. Results for adult SST neurons are missing in the Figure and are also not mentioned in the text.

Legend Fig. 4: ISH detection is done with a cRNA probe and not with an mRNA probe.

Detection of SST (by ISH) and NPY (by immunolabeling) neurons should be mentioned in the Results.

Results, page 10:

“Persistence of CRs has also been described in epilepsy-associated disorders in patients, based on histological studies 15-17” This statement is simply wrong, since there are many studies showing a loss of CR cells in the epileptic human hippocampus and in rodent epilepsy model.

“Although the onset latency was the same for animals that underwent convulsions in both groups (responders), the status epilepticus was longer in BaxCKO mutants (25.06 ± 7.382 min) compared to

controls”.

This sentence is confusing, since in Fig. 6C the latency is significantly reduced. Similarly, in the Legend of Fig. 6C it is stated that “Latency to develop seizures is increased” but the Histogram shows the opposite.

MEA recordings: please indicate the age of the animals

Discussion:

The discussion part about epilepsy is highly speculative and should be tuned down.

“Finally, CR released molecules such as reelin, could predispose to a higher susceptibility to lethal seizures in adults as a potential second trigger in late-onset epilepsy”.

Why that? There is no evidence that reelin is proepileptic. There are many publications showing that a loss of Reelin-expressing cells in the hippocampus accompanies the development of Ammonshorn sclerosis in TLE. Which is rather the opposite of the above mentioned sentence.

Material and Methods:

In general, this referee suggest to list the different method parts in the order the results are presented.

This means to put the very detailed description of the behavioral test in front etc.

I also think that, overall, the Methods part could be shortened and streamlined. In addition, one gets the impression as if the different experimental parts have been copied and pasted from elsewhere

Be consistent in using capital letters in the headlines.

Specifically:

Brain Harvesting and Dissection:

Correct: Xylasine to Xylaxine and Rampun to Rompun

In situ Hybridization:

Replace mRNA probes by cRNA, since that is the nature of the probes.

Many abbreviations are undefined in the text.

Was hybridization really performed at 70°C? How long were the probes? Which temperatures have been used for the washing steps?

Image acquisition and cell counting:

“Number of CRs or different subtypes of interneurons were counted using the ImageJ software, in the different hippocampal regions (identified based on histology) for each age and genotype.”

Please describe the counting procedure in more detail: which ROIs, how many sections/animal etc?

Patch-clamp recordings and analysis.

Why were adult C57BL6J mice, but not mutant animals used?? This looks like copy and paste ..

Correct the chemical formulas.

Behavioral tests:

I suggest including a table where the different tests, age of testing, and sexes are listed. It is very cumbersome for the reader to find these details in the text and to get an overview.

Kainate injection and recording:

“The latency, duration, number of crisis were recorded by a user blinds to the animals”.

What exactly was recorded? I guess behavioral signs of kainate-induced status epilepticus.

Please be more specific and describe in more detail.

RESPONSES to REVIEWER COMMENTS (in blue our responses to each comment)

Reviewer #1:

The manuscript by Riva and colleagues investigates the consequences of the aberrant survival of Cajal-Retzius cells (CRs) on hippocampal-dependent behaviors, network oscillations, and susceptibility to seizures. The authors take advantage of a previously established animal model with the invalidation of the pro-apoptotic factor Bax, which allows the unphysiological survival of CRs.

Results show that the aberrant survival of hippocampal CRs is associated with network alterations at the structural and functional level, cognitive deficits, and an adult epilepsy-prone phenotype.

In general, the work is very nice and addresses an important question with high translational impact because the persistence of CRs has been reported in different pathologies. Furthermore, CRs are still a relatively mysterious neuronal type, and therefore this work has the potential to significantly advance the field. There are, nevertheless, points that require improvements.

We thank the reviewer for considering our work very nice and addressing an important question with potential to significantly advance the field and high translational impact.

1) The discussion is too descriptive, does not address potential mechanistic scenarios, and does not clarify apparent inconsistencies in the results.

We thank the reviewer for prompting us to improve the discussion.

In more detail:

a) Are the effects observed in mutant mice caused by increased reelin availability due to CR survival or by computational effects driven by increased glutamatergic output due to rescued CRs? The increased complexity of pyramidal cell dendritic trees would suggest a contribution of reelin-mediated signaling. Is anything known about reelin levels in the hippocampus of the BaxCKO mouse? Could these be measured and compared to control animals? Regarding computational effects, have the authors attempted optogenetic stimulation of rescued CRs to evaluate changes of EPSCs in interneurons (as previously reported in the neocortex, ref 29)? These points should be at least discussed.

Q1: We thank the reviewer for raising these questions and prompting us to provide additional data and improve the discussion on the potential mechanistic scenario. The known role of Reln in regulating dendrite outgrowth and spine density is indeed consistent with the increased apical dendrites length and spine density observed in *BaxCKO* mice. Nevertheless, we have now tested Reln levels in *BaxCKO* and control mice using Western Blotting (new data added in **Fig. S3**) and we found no difference in the levels of Reln proteolytic products, suggesting no changes in released Reln in the extracellular space between conditions. These data point towards other mechanisms being at play. Moreover, the behavioral phenotypes observed might be associated with increased glutamatergic output of rescued CRs as it has recently been shown that altering this output can generate memory defects in adult mice (Anstötz et al., 2022). Accordingly, we have recently discussed the potentially differential effects of Reln and glutamate release from adult CRs (Sánchez-Bellot and de la Prida, *Epilepsy Currents* 2022). In this paper, we suggest that Reln defects would have a major impact on morphological traits and spontaneous seizure propensity, whereas CR glutamatergic release is rather posed to control microcircuitry, population activity and oscillations leading to changes in behavior. It is likely, thus, that the microcircuit integration of persistent CRs and their glutamatergic output supports the transition to circuitry susceptible to seizures upon second-hit insults. Therefore, we cannot exclude an interplay between both mechanisms to explain our observed phenotype. We have included these

ideas in the revised Discussion (pg. 16). Indeed, in the mouse model described by Anstötz et al. (2022), glutamate release by CRs is inhibited, leading to an increase in anxiety behavior and changes in novel object recognition, but not in novel object localization (Anstötz et al., 2022). The comparison of the BaxCKO and this latter model suggests that either an increased presence or reduced output of CRs may be sufficient to impair circuits underlying memory processing, although with non-overlapping effects on anxiety and NOL. This has been discussed in the revised version of the manuscript (pg. 15).

Regarding the computational effects to be tested via optogenetic stimulation of rescued CRs, as the reviewer mentioned, we saw no responses during patch-clamp or extracellular recordings of interneurons in the neocortex using light-evoked activation of ChR2 (Riva et al., 2019). This could be due to the low proportion of CRs compared to pyramidal neurons that would prevent the analysis and/or that direct synaptic inputs from CRs cannot account for the robust morphological and functional changes induced in pyramidal cells by the aberrant CR survival. This might be different in the hippocampus whereby Quattrocchio and Maccaferri (2014) have been able to record some paired synaptic activity in $Wnt3a^{Cre};ChR2$ possibly due to the higher density of CRs in the hippocampus. While synaptic connections between GABAergic interneurons to CRs are experimentally relatively easy to find (Quattrocchio and Maccaferri 2013), the degree of connectivity observed between CRs to postsynaptic partners is much lower (Anstötz et al., 2016) and other laboratories have failed to identify them (Marchionni et al., 2010). These experiments are, thus, challenging and their interpretation complex. During the pandemic we had to eliminate our mouse colony of the quadruple transgenic ($\Delta Np73^{cre/+};Bax^{lox/lox};Bak^{-/-};ChR2^{lox/+}$) and we feel that the time and costs necessary for producing this line again does not permit us to do this experiment. We hope in this reviewer understanding. We have, nevertheless, as suggested by the reviewer, discussed the potential circuit effects of persistent CRs throughout the revised version (pg. 14-16).

b) Some of the behavioral changes observed in mutant mice are very similar to what was recently reported in mice with inactivated glutamatergic functions of CRs (ref 28). This suggests that either loss or gain of function of hippocampal CRs are sufficient to impair circuits underlying memory processing. Could this consideration please be included in the discussion?

Q2: We agree with the reviewer and indeed our data together with the recent findings from Anstötz and colleagues (2022) suggest that both loss- or gain-of-function of hippocampal CRs is sufficient to induce deficits in spatial memory. However, Anstötz and colleagues show that a decrease in CR glutamatergic activity also promotes an increase in anxiety, alters spatial memory but leaves recognition memory intact, while we observe that increasing CRs number drives defects in spatial and recognition memory without affecting anxiety related behaviors in our mutant animals. We have added sections regarding these differences and similarities on pages 15 and 16.

c) Spine density increase in pyramidal cells with no parallel changes in spontaneous or miniature EPSC frequency. How do the authors interpret this apparent inconsistency? Do they think that the extra spines do not receive synapses and/or remain at an immature stage? Also, this seems to contrast with what was found in the neocortex of the same mutant mice, where a parallel increase in spine density and sEPSC frequency was observed (ref 29). This point should be explicitly mentioned and discussed.

Q3: We agree with the reviewer that is a point that needs further discussion. It is possible that the extra spines are still in an immature state. However, we think that a more parsimonious explanation for the inability to detect alteration in sEPSCs or mEPSCs in CA1 pyramidal neurons of mutant mice is most likely linked to the topographical distribution of the new spines

along CA1 pyramidal cells. In neocortical L2/3 cells of mutant mice we did observe an increase in eEPSCs in juvenile mice, but sEPSC were not tested. Thus, we cannot exclude that the eEPSCs could be altered also in the adult hippocampus. Nevertheless, in the neocortex this increase in eEPSCs was followed by an equal increase in the density of spines in both apical and basal dendrites (Riva et al. 2019). In addition, L2/3 pyramidal neurons of mutant mice showed more elaborated apical as well as basal dendritic arborizations. In contrast, in adult CA1 pyramidal neurons we now observed alteration in complexity of only apical dendrites and not basal. Also, the increase in synapse density is more pronounced in apical dendrites (estimation was made in dendritic segments located in SLM) when compared to basal. In our electrophysiological recordings, the detected spontaneous (or miniature events) synaptic activity is strongly biased towards events occurring close to the recording site (soma) and might not capture functional differences from distal parts of the dendritic tree. Thus, we think that the preferential difference in spine density in apical dendrites that present longer distances from the soma are just not captured by our standard sEPSCs or mEPSCs measurements. This possibility has now been added into the discussion (pg. 14).

d) Analysis of network oscillatory patterns. Firstly, the decrease in theta power in mutant mice appears to be selective for SLM recording sites, in contrast to gamma power, which was found to increase both in SLM and SP. What is the explanation for this selectivity?

Q4: We thank the reviewer for this important comment. Please note that entorhinal inputs at the SLM dominate during theta activity as they represent a major oscillatory drive in the dorsal CA1. Instead, gamma oscillations are brought about by local GABAergic microcircuits. Hence, there is spatial dissociation between the theta and gamma oscillations. We now refer to this point by providing supporting citation (Laszotczy and Klausberger, 2014) in the corresponding Results section pg. 7.

Secondly, it is unclear how the correlations (or lack thereof) between theta/gamma power and CR counts should be interpreted (Fig 3E). The data show a significant correlation only for theta power recorded in SP (but not in SLM). The lack of correlation between SLM theta and CR counts is counterintuitive and difficult to understand, given the observed differences between the power of SLM theta in control vs. mutant mice (which is presumably due to the different numbers of CRs).

Q5: We now include quantifications of CR density from control animals using immunostaining against P73. We also managed to identify additional probe tracks from some Bax mutants. New data confirm significant correlation between gamma oscillatory power at SP and CR density, as well as with theta attenuation at SLM. This matches well with the underlying physiology of theta (input generators at SLM) and gamma (local GABAergic perisomatic inputs at SP). See updated Fig. 3E,F and the Results section.

Lastly, given the significant correlation between SP theta power and CR counts wouldn't one expect to see a difference in SP theta power between control and BaxCKO mice (larger population of CRs)? Please clarify.

Q6: We thank the reviewer for this comment, which inspired the counting of control mice. We feel this is now clarified in the updated figure.

e) The methods section indicates that patch-clamp recordings were obtained from adult animals (P100-P120) and Fig 4 shows that enhanced survival of interneurons in mutant mice is specific for juvenile, but not for adult animals. How can a transient increase in interneuron survival (at juvenile stages) explain the increased frequency of sIPSCs and theta-gamma rhythmopathies found in adult mutant animals? As the interneuronal

population in adult mutant animals (apparently) does not differ from control mice, the most logical explanation would seem to be an increase in the excitatory drive from (an augmented population of) CRs onto interneurons. This point should be explicitly discussed.

Q7: We would like to thank the reviewer for pointing this out. We concluded that a transient increase in interneurons survival could permanently alter the hippocampal circuits thus explaining the behavioural defects and theta-gamma oscillatory power observed in adults. Certainly, our results open many possibilities and we can only speculate on at this point. As it has been shown that interneurons receive input from CRs it is possible that an increased number of these glutamatergic neurons could determine an enhanced recruitment of interneurons that will in turn explain the increased sIPSC observed on pyramidal neurons. This is in line with the observed augmentation in gamma oscillatory power, as increased population inhibition can lead to rebound excitation, putting an oscillation in motion.

It is also possible that the increase of inputs onto local interneurons by CRs could transiently reduce the amount of synaptic pruning and interneuronal cell death. The increased input onto interneurons may well continue into adulthood, evidenced by increased sIPSC frequency. We now mentioned these points in the discussion of the revised manuscript (pg. 14).

f) The only significant functional alteration reported in hippocampal pyramidal neurons of mutant mice seems to be an increase in spontaneous inhibition, whereas both intrinsic excitability and spontaneous excitation remain unchanged. This would suggest an overall decrease in the E/I balance, which is usually associated with a reduced probability of generating seizures and would seem in contrast to the observed lowered seizure threshold. How do the authors explain this apparent inconsistency? Were evoked IPSCs and EPSC affected in mutant mice? This point should be ideally tested experimentally or at least discussed.

Q8: The reviewer is indeed correct that at first sight an overall decrease in the E/I balance is generally expected to increase probability of seizures. Please note that, importantly, changes in E/I balance cannot be taken to be the sole cause for epileptic activity. Indeed even a decrease in E/I balance may lead to seizures as some seizures are accelerated by rebound firing following synchronous IPSPs. However, we think that our inability to detect alterations in EPSCs is linked to the fact that with electrophysiological recordings we mainly detect alteration in synapse density indiscriminately from dendritic segments close to the soma (see Q3, Reviewer 1). Yet, morphological analysis of CA1 pyramidal cells in mutant mice revealed selective alterations in apical dendrites that present longer distances from the soma. Multiple studies have illustrated that excitatory neurons are quite efficient at adjusting E/I balance (*e.g.* Xue and Scanziani et al., 2014). We think that the increased sIPSC frequency could be in fact a compensatory mechanism for the increased number of spines in an attempt to normalize pyramidal cell activity. *In vivo*, to avoid epileptic seizures both excitation and inhibition need to be balanced in a temporally precise manner, which we did not explicitly address in this study. We hypothesize that under kainate application the increased overdrive of apical dendrites of CA1 pyramidal cells (*e.g.* by increased activity of entorhinal inputs that target distal dendrites of CA1 pyramidal neurons) together with the increase number of synapses in such distal dendritic segments is at the origin of the observed epileptic seizures. One additional possibility is that the increased number of spines both on apical and basal dendrites are silent and expressing mainly the NMDA receptor so that in basal conditions no changes in excitability are observed. Nevertheless, when a second hit (kainate) is given, they become active and thus the E/I balance is moved toward excitation determining the lethal seizures. The increased sIPSC frequency could be a compensatory mechanism for the increased number of spines which

increase the inhibition of the system in basal conditions. We now comment this point in the revised version of the manuscript (pg. 16).

g) The significance of the experiments in Figure 7 is rather unclear. Are the authors suggesting that seizures originate in the entorhinal cortex and, from there, they then propagate to the hippocampus? A comparison of the activity in the different channels should be able to identify the site(s) of initiation of the seizure-like activity. Could this point please be addressed, and results discussed?

Q9: We thank the reviewer for this comment. Limbic seizures involve the entorhinal-hippocampal circuit, which is a hallmark of mesial temporal lobe seizures, such as those induced upon kainate treatment. The exact origin can flip around the entorhinal-hippocampal systems. The point here is not really where the seizure-like event originates in the reduced *ex vivo* preparation, but to support the functional effects of persistent CRs in biasing excitability using MEA. Following this question, we now report the site of origin in our *ex vivo* experiments and report *ex vivo* the differences in site initiation between control and mutants, supporting a more hyperexcitable DG in the latter. We now further report this point in the revised manuscript (pg. 11, 12).

2) Figure 1A. The upper left and right panels are identical. Please correct. Also, it is unclear why data for control animals are reported exclusively as a plunger plot, whereas individual results are also displayed for mutant animals. For consistency, please show individual results for both groups. In the text, it is stated that this experiment was performed in 2 assays at 2 min distances, but it is unclear what is actually represented in the plot (1st assay only, 2nd assay only, average of the two assays). Please clarify.

Q10: We thank the reviewer for pointing to these mistakes. In the revised manuscript, we have now corrected Fig. 1A and the text (Material and Methods section and figure legend), provided individual results for both groups, and homogenized the graphs in all figures. Please note that boxplots have been used in Fig. 3 as the scale of changes is small.

3) Figure 1A. Bottom right panel. The latency to find food tested at a 24-hour interval is reported as not significant for male mutant mice. However, these data are very similar to what was found for male control animals. Are the authors sure of this result? In particular, it is unclear what statistical test was used, as in the text it is mentioned that a two-way ANOVA with post hoc Sidak's multiple comparison test was used, whereas in the legend it is stated that a Mann-Whitney test was used. As this was a paired comparison (1st vs 2nd trial in the same animals), shouldn't a Wilcoxon signed-rank test be used? Would this test result in significance?

Q11: We thank the reviewer for pointing this out. We now provide the graph on the same scale than the controls. This change allows to see better the difference in learning between day 1 and 2 for the control animals compared to the mutants. We also thank the reviewer for the point made about the statistical test. We initially used a RM-two way ANOVA, but wrongly written Mann Whitney in the figure legend. We have now performed the more appropriate Wilcoxon signed-rank test. Our results are still significant following this novel statistical test and are presented in Figure 1A. In the Table S2 we provide the results for both the Wilcoxon signed-rank test and the RM-two way ANOVA, as this has been suggested by Reviewer 2. Both tests show a significant impairment in mutant animals.

4) Figures 2B (and 4D). It is very hard to see CRs (interneurons) on printouts. Could the authors please either increase the contrast or include insets at increased magnification?

Q12: We thank the reviewer for this suggestion. We have now added insets at high magnification in the revised Figures 2 and 4.

5)Figure 3. The vertical size of the figure is rather small, which makes it difficult to see the data. Also, the font size of the axis labels is too small to read. Please increase the size.

Q13: We thank the reviewer for this point. We have now amended the Figure 3 as suggested.

6) Figure 6. Please include examples of EEG recordings.

Q14: As mentioned in the Material and Methods section, Kainate injections in Fig. 6 were not followed by *in vivo* recording, but recorded via a CaptureStar software for 2 hours. The latency, duration, number of crisis were recorded by a user blinds to the animal genotype. We have now further clarified this point in the revised manuscript (pg. 31, 32).

7) In general, the consistency of the illustrations should be improved by using the same type of plot and convention. For example, could the authors please use either plunger plots plus individual results (as in Figs 1B, 2, 4, etc..) or mean and error bars plus individual results (as in Figs 1C, 1F, 5C, 5D, etc..) or boxplots and individual results (Fig 3) throughout the paper? Also, could the same convention to distinguish control (white bars and empty circles) from mutant mice (blue bars and black circles) please be used for every figure?

Q15: We thank the reviewer for this suggestion. As requested, we have now homogenized all figures in the revised manuscript.

8) Page 3. “These cell populations are unique as they completely disappear at the end of cortical maturation shortly after birth in mice”. This statement (referring to CRs in general) is incorrect as hippocampal CRs persist in adult animals and never completely disappear. Please reword this sentence.

Q16: We apologize for this mistake and the lack of clarity in the text. We have now corrected this sentence (pg. 3) in the revised manuscript.

9) Page 3. “Strikingly, transient hippocampal CRs stay longer in juvenile mice and yet a small 15% remain integrated in adult microcircuit”. This sentence is inherently contradictory as it claims that “transient” neurons “remain in adult microcircuits”. Please correct. Also, I am not sure how the 15% was calculated as none of the references quoted (refs 27, 28) address quantitatively this specific point. Reference 37 shows that from P7 to adulthood a ~30% of hippocampal CRs persist. Please modify.

Q17: We agree with the reviewer that the references 27 and 28 were not correct. Indeed, in reference 37 it is shown a 30% increase of hippocampal CRs when the animals are exposed to enriched environment. Our phrase was referring to the decline in number of CRs that occur with age as in our case we are preventing cell death by blocking apoptosis of CRs. Our interpretation of the 15% is based on the work of Anstötz and colleagues (2016) in which the authors observed a 85% decrease of CRs cell density near the HF from P8 to P60. However, Anstötz and colleagues (2019) also showed that at P7 they have a CR density of 17.33 and 6.74 at P60 which is approx. 38% of CRs left. In both cases animals are Tg(CXCR4-EGFP), who also label interneurons and GFP expression is dependent on Cxcr4 expression at any given time. We have worked with DeltaNp73^{Cre} and Wnt3a^{Cre} mouse lines allowing permanent tracing that are also described in Anstötz et al. (2018). As requested, we have now changed the sentence “Strikingly, transient hippocampal CRs stay longer in juvenile mice and yet a small 15% remain integrated in adult microcircuit” in the text (pg. 3).

10) Page 4. “A primary screen of 65 tests [...] was performed”. However, only a much more limited number of tests is presented in Supplementary Table 1. If 65 different tests were performed, then these results should be included in a Table.

Q18: As requested we have added a new Table including all of the 85 tests performed in mutant mice and control littermates for both males and females.

11) Data presentation. Could the actual data values please be included either in the text or in a supplementary table? Most of the time, results are presented in the text only by indicating the statistical test that was performed and its probability. Also, it is often quite difficult to find the exact sample size (n) and unit of measurement for each experiment. Could the authors please consistently include these with the numerical measurements, test performed, and probability (either in the text or in a supplementary table)?

Q19: As requested, for each experiment performed in this study we have now included the exact sample size (n) and unit of measurement together with the numerical measurements, test performed, and probability in a supplementary Table (Table S2).

Reviewer #2:

In this interesting and very thorough manuscript, the authors utilize a genetic mouse model of Bax in activation in p73 expressing cells to suppress programmed cell death of Cajal Retzius (CR) cells derived from the septum and from cortical hem. The authors use this model to assess the effects of CR persistence on behavior, and cellular and circuit activity in the hippocampus. They also report impairment in hippocampus-dependent behaviors, specifically spatial and episodic-like memory in adults, but not juveniles. These behavioral changes were associated with increases in theta oscillations, but decreases in gamma activity. There was also a decrease in NPY-GABAergic cells in juveniles and a changes in pyramidal cell spine density in adults. Mutant mice also had increased likelihood of tonic-clonic seizure activity.

In general, this was an excellent manuscript. There is an exhaustive list of behavioral tasks performed and the results range from behavior to circuit activity to neural pathology. The results contribute enormously to our understanding of the role of transient CR cells in hippocampal development and function and demonstrate that their demise via programmed cell death is as important to normal function as their transient role in establishing circuit connectivity. However, there are a few concerns with the manuscript that should be easily addressed.

We thank the reviewer for considering our work interesting, very thorough, excellent and using an exhaustive list of experiments contributing enormously to our understanding of the role of transient CRs.

Q1: Can the behavioral tasks performed be disentangled from cortical function? Presumably, the septum CRs in cortex are spared in this mouse model? For example, novelty detection relies on hippocampal/mPFC circuits whereas object recognition relies heavily on lateral entorhinal cortex (LEC) and upstream perirhinal connectivity with PFC. Deficits on this task suggest impaired LEC processing. Perhaps CR survival in regions “upstream” of the hippocampus is contributing to the observed effects as well? The addition of some language to address this possibility, or at least toning down the claims that the behavioral deficits are definitely and solely due to CR persistence in hippocampus is warranted.

We would like to thank the reviewer for pointing this lack of clarity in the manuscript. Indeed, we cannot conclude that all altered behaviors of our mutant mice are solely hippocampal-driven,

but can also be a result of alterations in other neocortical regions. However, it is important to mention here that we have performed several behavioral tests that are known to assess selectively hippocampal-dependent behavior, such as the Novel Object Location (NOL) and the Morris Water Maze (MWM) tests, and in which our mutant mice display severe deficits. Therefore, although we cannot conclude that all behavioral deficits are a result of hippocampal alterations, these series of behavioral tests together with the rhythmopathies and MEA alterations observed in the hippocampus strongly argue in favor of a major contribution of an impairment of hippocampal circuits in our phenotype. We have now clarified this point in the revised manuscript and toning down our claims suggesting that the behavioral deficits are “solely” due to CR persistence in hippocampus.

We agree with the reviewer that the entorhinal cortex is a very important region and it possibly involved in the phenotype. Indeed, CRs localized in the SLM and OML were shown to be an important temporary target for entorhino-hippocampal axons (Del Rio et al., 1997; Soriano et al., 1994; Super and Soriano, 1994). CR ablation indeed determines an off targeting of these afferences (Del Rio et al., 1997). As in the mutant BaxCKO mice an enhanced number of CRs is found in CA1, it is possible that their persistence could interfere with the ingrowth of the entorhino-hippocampal afferences resulting in incorrect circuit maturation and the observed behavioral defects. However, lesions of the medial entorhinal cortex were shown to lead to defects in spatial memory in the MWM, but not in paradigms such as NOR and CFC (Hales et al., 2014, reviewed in Canto et al., 2008) and, conversely, those of the lateral entorhinal cortex in associative memories (Wilson et al., 2013). Moreover, alterations of entorhinal-hippocampal connections appear to influence also the DG, in addition to CA1/CA3, where we see no major defects in BaxCKO mutants. Finally, the correct laminar organization of both the alvear and the perforant pathways were shown not to be dependent on the number of GABAergic neurons in the hippocampus (Pleasure et al., 2000; Super et al., 1998). Lastly, stimulation of entorhinal-hippocampal connections leads to reversion of epileptic seizures in kindling and pilocarpine models of epilepsy (Xu et al., EBioMedicine 2016) contrary to what would be expected in BaxCKO, whereby susceptibility to seizures is enhanced (see below). Together, these results strongly argue against a simple effect of CRs and interneuron survival in CA1 on the entorhinal-hippocampal system as mediating the behavioral defects observed in BaxCKO mutants. We cannot speculate extensively on this subject at this point due also to space constraints and criticisms from another reviewer of too much speculation in the Discussion, but we have maintained the possible EC connection in the revised discussion section (pg. 15, 16).

Q2: There was no effect on contextual fear conditioning which is hippocampus dependent including CA1. Can the authors address why they think this behavior is normal despite their other neural and behavioral findings?

Lesions of the rodent hippocampus invariably abolish context fear memories formed in the recent past but do not always prevent new learning. A series of studies indicate that the hippocampus is normally involved in context conditioning but not always necessary for learning to occur (Wiltgen et al., 2006). Different networks could be involved in the control of fear memory (Jimenez et al., 2020). Although the hippocampal–amygdala pathway has been implicated in the retrieval of contextual fear memory, the mechanism by which fear memory is encoded in this circuit remains still largely elusive.

Recent studies suggest that it is mainly the ventral CA1 (vCA1) hippocampal projections to the basal amygdala (BA), paired with aversive stimuli, that contribute to encoding conditioned fear memory. We can hypothesize that the presence of CRs in the hippocampus do not alter these specific pathways. Importantly, aversive fear memories are also dependent on the amygdala, which seems not to be affected in the BaxCKO mutants as no DeltaNp73 –derived cells populate

this region. This point and the absence of phenotype in contextual fear conditioning has been developed in the discussion section of the revised manuscript (pg. 5, 6, 14, 15).

Q3: In Figure 1A, bottom panel: the statistical analyses are performed separately for controls and mutant mice, demonstrating that there is a significant decrease in latency between Trials 1 and 2 in controls, but not in mutant mice. However, the Y axis are different on the control and mutant mice graphs. It appears that mutant mice are faster on Trial 1 than controls, but do not really differ much in the latencies on Trial 2, suggesting that there is no significant decrease in latency in mutant mice because they are already performing quickly in Trial 1. This leads to a different conclusion than the authors make. The suggestion is to conduct a two-way (Trial X mutation) ANOVA to assess this issue.

We thank the reviewer to have indicated the problem with the presentation of the Y axis with different scales, which was inducing confusion and a distinct interpretation. We have homogenized the Y axis for mutants and controls in lower panels of Fig. 1A. This shows that controls are more variable than mutants. However, Fig. 1A upper left shows that there are no differences in the latency in Trial 1 between controls and mutants. Still the controls are faster in finding the food between the two trails when compared to the mutant mice. So we believe that our conclusion is now more clearly supported and the revised figure is now included in the revised manuscript. Moreover, we have now performed a Wilcoxon signed-rank test (as suggested by Reviewer 1), but we have also included the results of a RM two-way ANOVA in Table S2. Both tests still point at a defect in learning in mutant animals.

Q4: On page 6, the authors report that they were unable to discriminate between the SLM of the CA1 and the outer molecular layer of the dDG. They go on to say that because no differences were found in the vDG, the surviving CRs must be in CA1 SLM. This is not a sound conclusion as dorsal and ventral DG (i.e., infra- and supra blades of the outer molecular layer) can have different functions. Also, the explanation about combining the CA1/CA2/dDG should be included in the caption of Figure 2. As is, it appears that CRs are being counted in CA1 and CA2 proper (not in SLM), but clearly there are no β gal cells in the pyramidal cell layer of those regions.

We agree we the reviewer that we cannot conclude that the surviving CRs are specific to the CA1 regions if we cannot discriminate the CA1 SLM from the dDG OML. CRs are only present in the HF and not in the pyramidal layer (or hippocampus proper as mentioned by the reviewer). We have now replaced with quantifications along the HF and vDG. We also removed the conclusion that the reviewer was referring too. In Materials and Methods we have more clearly specified how the cells were quantified (pg. 26).

Q5: On page 10, “CRs persistence leads to enhances susceptibility to seizures”: This paragraph refers to data from analysis of video-EEG recordings. Are these data shown in the manuscript?

This paragraph reports of data based on EEG recording in juveniles showing no spontaneous seizures and on video-recording via a CaptureStar software for 2 hours in juvenile and adults after kainate injections. The latency, duration, number of crisis were recorded by a user blinds to the animal genotype. We have now clarified the methodology (pg. 31, 32) and provided a Figure S6A showing examples of EEG recording in juveniles with no abnormal paroxysmal activity.

Q6: In Figure 6A and 6C: The latency to seizure did not differ between control and mutant responders. However, Figure 6A shows a significant difference in latencies (180 vs 24). Does this mean that non-responders were assigned the maximum latency to get the results

shown in 6A? This seems to already be covered in the seizure incidence data and creates some confusion. It should be explained more clearly in the figure caption and/or results or latency in non-responders should be removed.

We thank the reviewer for pointing out this problem. As mentioned by the reviewer, the non-responders were assigned the maximum latency for results shown in Fig. 6A. In Fig. 6C the left panel shows the latency to develop seizures for all animals and the right panel only for the responders (n=6 for controls and n=15 for mutants). We have now provided a clearer explanation in the figure legend and results.

Q7: On page 12, reference 49: Based on the title of the reference this study examined a progesterone receptor antagonist and not estrogen treatment. The authors should check this.

We thank the reviewer for bringing this to our attention and have corrected it accordingly.

Q8: In general, the manuscript should be checked for grammar and typos. We have now corrected the grammar and typos in the revised manuscript.

Reviewer #3:

The present manuscript entitled “Aberrant survival of hippocampal Cajal-Retzius (CR) cells leads to memory deficits, gamma rhythmopathies and susceptibility to seizures in adulthood” by Riva et al. builds on previous work of the same group investigating the persistence of Cajal Retzius cells (CR) in the neocortex. CR normally undergo programmed cell death (PCD) after completion of neocortical networks. The results of this study were obtained in mouse mutants (BaxCKO Δ Np73Cre = BaxCKO) in which Bax-dependent PCD of a CR subtype is prevented. A battery of behavioral tests was performed with these mutants (males and females) and compared to wild type animals. Authors found in particular that hippocampus-dependent cognitive functions were normal in young adult, but affected in adult mutant mice of both sexes as shown by a Cookie test, Morris water maze, novel object recognition and object location tests. Authors concluded that the persistence of transient CR may induce a major impairment of hippocampal function. Therefore, they investigated the pattern of CR cells in their mutant mouse line and found an increased survival of LacZ-expressing cells (= CR cells) in the adult hippocampus. Based on these observations they conducted in vivo LFP recordings with 16-channel silicon probes and in vitro patch-clamp and a multi-electrode array (MEA) recording to determine potential consequences of CR cell survival. They found an attenuation of theta oscillations and enhanced gamma activity in the dorsal CA1 region, transient changes in the number of NPY+ GABAergic cells in juvenile mice and alterations of CA1 pyramidal cells spine density in adults, enhanced frequency of inhibitory currents in adult hippocampal pyramidal cells and an increased susceptibility to kainate-induced status epilepticus.

Results of this study are basically interesting. Several state of the art techniques (behavioral tests, immunocytochemistry, in situ hybridization, patch-clamp recordings, multi-electrode array (MEA) recording and in vivo LFP recordings with 16-channel silicon probes) have been applied but this referee has concerns regarding the nature of the surviving hippocampal neurons in BaxCKO mouse mutants. The aberrant survival of CR cells into adulthood is the basis for all follow-up experiments in this manuscript and should be more convincingly demonstrated. See specific comments and concerns below.

This reviewer major concern is whether the surviving cells are CRs. This implies that she/he puts into questions the use of genetic tracing to follow the life of cell populations. In general,

the expression of P73 and the use of Cre lines specific for CRs, namely the ones used in this study (DeltaNp73^{Cre} and Wnt3a^{Cre}), have been validated in numerous publications from distinct laboratories in both the neocortex and the hippocampus (among which Anstötz et al., 2022; Tissir et al., 2009; Ledonne et al., 2016; Riva et al., 2019; Meyer et al., 2022; Quattrocchio and Maccaferri, 2014; Anstötz et al., 2018, Louvi et al., 2007; Gu et al., 2011). We have now confirmed that all traced cells surviving in the HF are CRs by providing independent quantifications using 3 molecular markers, which together unequivocally identify CRs in addition to genetic tracing with two distinct reporters.

Q1: Persistence of CR cells in TLE with AHS as stated in the introduction is controversial, there is only one study reporting an increase, whereas many papers demonstrate a loss of CR cells in human patients as well as in rodent epilepsy models.

We have clarified this point in the introduction and mentioned that there is a controversy in place (pg. 3). We would also like to mention that CRs loss or increase in epilepsy is also difficult to interpret as in many cases seizures have already occurred and thus the observed phenotype might not necessarily be the cause but rather the consequence of epilepsy. Our model unequivocally shows that the phenotype is due to CR aberrant survival prior to the occurrence of seizures. Moreover, we refer to 4 distinct studies in different pathologies.

Q2: In the manuscript, three different age groups (P24, 9 weeks and 3 months) have been analyzed. This should be indicated clearly and consistently in the Figures and Legends. It is quite tedious and confusing to get it in the present manuscript. In addition, there is quite a bit of redundancy in Legends, they could be shortened substantially.

We agree with the reviewer that the different ages can be difficult to follow. We have made sure all ages are indicated clearly and consistently in the Figures and Figure legends. In addition, we now added a table in which all data are represented regarding the age, gender, n= and the tests used (Table S2).

Q3: Morris Water Maze (MWM) test: is this really appropriate for mice, since they are afraid of water and consecutively very stressed? A dry land maze such as the Barnes would be more suitable.

We would like to thank the reviewer for pointing this out. We agree that the Morris water maze could be stressful for mice. However, it has been shown that the Barnes test is not the most appropriate spatial test for female mice (Gawel et al, 2019). In order to test spatial memory defects we have thus performed in addition to the MWM the Novel Object Location (NOL), a low-stress and efficient test that assess hippocampal-dependent spatial memory. The results observed in the NOL corroborate the one obtained in the MWM.

Q4: Fig. 1: Were the same mice tested in the different behavioral paradigms? Numbers of mice are different in the different tests.

We would like to apologize for the lack of clarity concerning the behavioral analyses. We have used different cohorts of mice. The sequence of the test for each cohort has now been added in the Materials and Methods section (pg. 18, 21).

Q5: Why only the males are shown in Fig. 1? Data on females are all put into the supplement, although they show also deficits in memory.

As the deficits were similar in both males and females and in order to not overload the figures, we decided to include the female data in the supplemental figures. However, we will be happy to invert if the reviewer finds it necessary, but unfortunately we cannot include all of the data for both genders in the main figures.

Q6: Concerning female mutants: only tests performed with 3-month-old mice are shown. Is the age-dependent switch from normal behavior in young adults to impaired memory in older ones also present in females? As far as I understood the plethora of behavioral tests, females were only tested at young age and in the adult stage.

We have done the primary screen on both juvenile males and females (now Table S1). Since there were no behavioral differences for both gender, and to follow ethical rules (3R) and reduce costs, we continued only with young adult males. Since we found defects in behavior at adult ages in male mice, we then returned to perform the same behavioral tests in adult females.

Q7: Fig. 2. What is juvenile and what is adult? In B it is stated P24 hippocampus, is this correct?

We thank the reviewer for pointing to this mistake. We have now corrected this point in the revised manuscript.

Q8: My main concern with this Figure is that the quality of the micrographs in B is poor. Labeled cells cannot be recognized. In addition, why should there be CR cells in CA3? CR cells are usually confined to the hippocampal fissure.

We apologize for the lack of details. We have tried to provide a view of the entire hippocampus. Please, note that the Tau locus expression is lower in adults. We now provide higher magnification panels as well as images with ROSA^{tom} as reporter which is brighter and more highly expressed in adult CRs. Regarding CRs in CA3, please note that CA3a,b, which are the distal sectors of CA3, also have SLM and are exposed to the fissure. So, CRs can be also found there. To avoid misunderstanding, we now removed counting in CA3 alone and show quantification in the entire HF (CA1/CA2/CA3 and dDG). The results remain the same and are now presented in the revised manuscript (Figure 2).

Q9: It is also surprising that the number of LacZ-positive cells is increased in CA3 in the adult stage when compared to juvenile animals. How can this be explained?

We thank the reviewer for this comment and we agree that this subdivision (relying on anatomical landmarks) is not appropriate for CR quantification in the hippocampus. Accordingly, with previous studies of CR distribution in the hippocampus (from many laboratories and in particular from the recent review by Anstötz et al., 2019) we now used quantifications along the HF. These confirm an increase in CRs in the HF of both juvenile and adult mutant mice with similar numbers of CRs using genetic tracing, molecular markers and a consistent decrease between juvenile and adults.

Q10: Did the authors verify the CR-specificity of the LacZ-expression with CR-markers? Calretinin or reelin could be used to label CR cells at the hippocampal fissure (see Anstötz et al., 2018 and others).

We have checked with multiple CR markers in addition to genetic tracing (as reported initially). We provide images and quantifications with ReIn, P73 and GABA, being ReIn+, P73+ and GABA- the signature of hippocampal CRs. These data are now presented in Figure 2D and 2E and unequivocally show that genetically labeled surviving cells are CRs.

Q11: Fig. S3. This referee does not understand the rationale of using BaxCKOwnt3acre mice. They express LacZ not only in CR cells but also in granule cells and pyramidal cells as obvious in Fig. S3.

We agree with the reviewer. This is indeed a Cre line that has been used before (Quattrocchio and Maccaferri, 2014; Anstötz et al., 2018, Louvi et al., 2007; Gu et al., 2011) to trace CRs derived from the cortical hem that represent all CRs in the hippocampus. However, it also shows that some pyramidal neurons derive from Wnt3a expressing territories are also present in the control animals. This is the reason why we did not use these animals for the behavioral or functional analyses. Nevertheless, this line allows us to trace hem-derived CRs and show that some of these hem-derived CRs survive in the HF of the hippocampus, while they do not in the neocortex (Ledonne et al., 2016; Riva et al., 2019).

Q12: Fig. 4. Results for adult SST neurons are missing in the Figure and are also not mentioned in the text.

Since the number of SST did not change in juveniles and that there were no differences in Gad1(Gad67)⁺ neurons in the adult, we did not analyze SST in adult mice in the original manuscript. As requested by the reviewer, we have now performed the analysis of SST (as well as NPY) cellular density in adults. No differences were observed. These novel data are now included in Fig. S5 of the revised manuscript.

Q12: Legend Fig. 4: ISH detection is done with a cRNA probe and not with an mRNA probe.

We thank the reviewer for this comment and we have now corrected this mistake in the revised manuscript.

Q13: Detection of SST (by ISH) and NPY (by immunolabeling) neurons should be mentioned in the Results.

We thank the reviewer for this comment and we have now mentioned in the Figure legend that the detection of SST⁺ neurons was obtained by ISH, while NPY⁺ neurons by immunolabeling.

Q14: Results, page 10:“Persistence of CRs has also been described in epilepsy-associated disorders in patients, based on histological studies 15-17” This statement is simply wrong, since there are many studies showing a loss of CR cells in the epileptic human hippocampus and in rodent epilepsy model.

We agree with the reviewer that the issue is controversial as different studies associate both persistence or loss of CRs with convulsive disorders (see response to Q1 of this Reviewer). However, we feel the sentence is not wrong. There are studies describing persistence of CRs in epileptic patients. As requested, we have rephrased this sentence (pg. 10). We would also like to respectfully mention that it is difficult to firmly conclude from studies showing loss of CRs in status epilepticus whether this loss of reelin-releasing cells is a cause or an effect of the observed seizures. What we have modeled in our mice is what happens when CRs do not die and the consequences thereof. Our results show for the first time that persistence of CRs promotes the phenotypes observed in our work from behavioral deficits to the susceptibility to kainite-induced seizures, rhythmopathies and MEA defects. Moreover, we have discussed the results of lack of Reln in pro-epileptogenic pathologies throughout the manuscript and in particular on pg.13. Last, we now discussed in more details (pg. 15-17) in the discussion of the revised manuscript the new emergent line of enquiry into the glutamatergic involvement of CRs in the local hippocampal adult microcircuitry.

Q15: “Although the onset latency was the same for animals that underwent convulsions in both groups (responders), the status epilepticus was longer in BaxCKO mutants (25.06 ± 7.382 min) compared to controls”. This sentence is confusing, since in Fig. 6C the latency

is significantly reduced. Similarly, in the Legend of Fig. 6C it is stated that “Latency to develop seizures is increased” but the Histogram shows the opposite.

We thank the reviewer for catching this mistake in the figure legend, which we have corrected.

Q16: MEA recordings: please indicate the age of the animals

We have added the age of the animals in the figure legend of the revised manuscript.

Q17: The discussion part about epilepsy is highly speculative and should be tuned down. “Finally, CR released molecules such as reelin, could predispose to a higher susceptibility to lethal seizures in adults as a potential second trigger in late-onset epilepsy”. Why that? There is no evidence that reelin is proepileptic. There are many publications showing that a loss of Reelin-expressing cells in the hippocampus accompanies the development of Ammonshorn sclerosis in TLE. Which is rather the opposite of the above mentioned sentence.

We have now discussed more extensively the possible role of Reln and which phenotypes could be linked to increased Reln in the discussion section of the revised manuscript. In addition, we have now included additional data showing that Reln is not increased in BaxCKO mutants (new Fig. S3).

Q18: In general, this referee suggest to list the different method parts in the order the results are presented. This means to put the very detailed description of the behavioral test in front etc.

As requested by the reviewer, we have now listed the methods in the order the results are presented.

Q19: I also think that, overall, the Methods part could be shortened and streamlined. In addition, one gets the impression as if the different experimental parts have been copied and pasted from elsewhere

We agree with the reviewer and shortened the Material and Methods section of the revised manuscript.

Be consistent in using capital letters in the headlines.

We are now consistent in using capital letters in the headlines.

Correct: Xylasine to Xylaxine and Rampun to Rompun

We have now corrected this mistake.

In situ Hybridization:

Replace mRNA probes by cRNA, since that is the nature of the probes.

We have now corrected this mistake.

Many abbreviations are undefined in the text.

We have now added definition for all abbreviations in the text.

Was hybridization really performed at 70°C? How long were the probes? Which temperatures have been used for the washing steps?

The hybridization and washes were performed at 70°C as was mentioned in the original version. The protocol is standard and has been used in many different studies.

Image acquisition and cell counting:

“Number of CRs or different subtypes of interneurons were counted using the ImageJ software, in the different hippocampal regions (identified based on histology) for each age and genotype.” Please describe the counting procedure in more detail: which ROIs, how many sections/animal etc?

We have now added a clear description of the counting procedure (pg. 26) and number of animals in Table S2.

Patch-clamp recordings and analysis.

Why were adult C57BL6J mice, but not mutant animals used?? This looks like copy and paste ..

We thank the reviewer for spotting this mistake, which we have corrected in the revised manuscript.

Correct the chemical formulas.

We corrected.

Behavioral tests:

I suggest including a table where the different tests, age of testing, and sexes are listed. It is very cumbersome for the reader to find these details in the text and to get an overview.

As requested by the reviewer, we have now provided a Supplementary Table (Table S2) where the different tests, age of testing, sexes, number of animals and significance are listed.

Kainate injection and recording:

“The latency, duration, number of crisis were recorded by a user blinds to the animals”.

What exactly was recorded? I guess behavioral signs of kainate-induced status epilepticus. Please be more specific and describe in more detail.

We have described the Kainate injection and recording procedure in more details in the Material and methods section of the revised manuscript (pg. 31, 32).

REVIEWERS' COMMENTS

Reviewer #1 (Remarks to the Author):

The manuscript has substantially improved, and my comments have been appropriately addressed. I congratulate the authors for their important work that will certainly be appreciated by the field.

My only remaining minor concern regards lines 15-17: "Unlike the majority of cortical neurons, CRs almost completely disappear by PCD during early postnatal weeks". This generalization is correct for the neocortex, but not for the hippocampus, and I suggest to the authors that they should modify this sentence. This is probably important because in the following lines (lines 21-22) they do specifically address the "Abnormal survival of CRs" linked to "impairment of hippocampus-dependent function", which a general reader may think was the cortical region referred to by the initial lines (15-17).

Reviewer #2 (Remarks to the Author):

The authors have been quite responsive to reviewers' comments and the manuscript is much improved. In particular, the discussion and some of the interpretations of results have been toned down and/or contextualized in important ways. Additionally, some methodological details have been clarified. Important improvements in graphical representation of data and statistical analyses have been made.

Overall, this was an excellent manuscript and the revisions have tightened up the manuscript in important ways.

Reviewer #3 (Remarks to the Author):

The revised manuscript of Riva et al. has improved a lot. All points of criticism from my side have been answered or implemented by the addition of new data or appropriate text changes.

I found a few inconsistencies in the revised version which are specified below:

Introduction

....Although still controversial, the persistence of CRs during advanced postnatal life is reported in different pathologies¹⁴. In particular, abnormal persistence of transient cells with molecular signatures of CRs are present in cases of temporal lobe epilepsy (TLE), Ammon's horn sclerosis, focal cortical dysplasia and polymicrogyria.¹⁵⁻¹⁷.....

¹⁶ Kostovic, I. & Judas, M. Transient patterns of cortical lamination during prenatal life: do they have

implications for treatment? *Neurosci Biobehav Rev* 31, 1157-1168 (2007).

This reviewer does not understand why Kostovic & Judas (2007) have been cited in this context, since this review does not cover the abnormal persistence of transient cells but the formation of early neuronal circuits in prenatal life.

Results:

Fig. 1 C, D: There is a mistake in labeling: in both Figures testing is used twice but should it not be training followed by testing? This also applies to Fig. S 2A,B.

Figure S3 B: is it not surprising that a 30% increase in Reelin-expressing CR cells does not lead to increased Reelin levels in the hippocampus (see Western blot data)?

Table S2: It is impossible to read the contents of this table.

Figure 3E: P73-positive cells were counted and are shown.

In the methods section "Electrode track verification and CRs quantification", however, it is indicated that TdTomato+ cells in the SLM were counted. What is correct?

Fig. S4: Why was SST expression determined by immunolabeling here and not by ISH as done in Figure 4?

... Although different studies associate both gain or loss of CRs with convulsive disorders, persistence of CRs has also been described in epilepsy-associated disorders in patients, based on histological studies 15-17....

Again ref. 16 is not appropriate for this statement, please cite here other papers.

Ref. 75 does not deal with Reelin deficiency in TLE.

POINT-BY-POINT RESPONSE (in blue) TO EACH REVIEWER COMMENTS

REVIEWER #1:

My only remaining minor concern regards lines 15-17: “Unlike the majority of cortical neurons, CRs almost completely disappear by PCD during early postnatal weeks”. This generalization is correct for the neocortex, but not for the hippocampus, and I suggest to the authors that they should modify this sentence. This is probably important because in the following lines (lines 21-22) they do specifically address the “Abnormal survival of CRs” linked to “impairment of hippocampus-dependent function”, which a general reader may think was the cortical region referred to by the initial lines (15-17).

We have changed the abstract as requested and clarified the differences between neocortex and hippocampus in CRs survival in adult mice.

REVIEWER #3:

The revised manuscript of Riva et al. has improved a lot. All points of criticism from my side have been answered or implemented by the addition of new data or appropriate text changes.

I found a few inconsistencies in the revised version which are specified below:

Introduction

....Although still controversial, the persistence of CRs during advanced postnatal life is reported in different pathologies¹⁴. In particular, abnormal persistence of transient cells with molecular signatures of CRs are present in cases of temporal lobe epilepsy (TLE), Ammon’s horn sclerosis, focal cortical dysplasia and polymicrogyria.¹⁵⁻¹⁷.....

¹⁶ Kostovic, I. & Judas, M. Transient patterns of cortical lamination during prenatal life: do they have implications for treatment? *Neurosci Biobehav Rev* 31, 1157-1168 (2007).

This reviewer does not understand why Kostovic & Judas (2007) have been cited in this context, since this review does not cover the abnormal persistence of transient cells but the formation of early neuronal circuits in prenatal life.

We thank the reviewer and the reference has been replaced.

Results:

Fig. 1 C, D: There is a mistake in labeling: in both Figures testing is used twice but should it not be training followed by testing? This also applies to Fig. S 2A,B.

We thank the reviewer for pointing out this mistake that now we have corrected in the revised version.

Figure S3 B: is it not surprising that a 30% increase in Reelin-expressing CR cells does not lead to increased Reelin levels in the hippocampus (see Western blot data)?

We think that although rescued CRs still produce RELN, the quantity of the protein in CRs appears much lower than in CRs at embryonic stages and if compared to that detected in RELN⁺ interneurons. This might be due to size difference between these two cell types (as it is visible in Figure 2E). Moreover, first the amount of Reelin protein detected by WB also depends on the percentage that CRs represent with respect to the total of other cell types (pyramidal neurons, glial cells etc...). We do not know what is this percentage but it is very low when considering the whole hippocampus (which was used for WB). Thus even a 30% increase in CRs number would represent a minimal cell number within the total cell number in the hippocampus. Second, Reelin is a secreted glycoprotein cleaved in the extracellular environment. Cleavage representing the active protein secreted by CRs or GABAergic neurons cannot

be properly quantified unless with single cell proteomic, which is not available at the moment. Nevertheless, our results using WB allows to conclude that there is not an overall higher ReIn expression in the hippocampus of Bax mutants.

Table S2: It is impossible to read the contents of this table.

We agree that it could difficult to read the table, due to the high amount of contents, however we have created this table in order to present all the information that the reviewers asked us.

Figure 3E: P73-positive cells were counted and are shown.

In the methods section "Electrode track verification and CRs quantification", however, it is indicated that TdTomato+ cells in the SLM were counted. What is correct?

We thank Reviewer 3 for spotting this issue. We have corrected and clarified the methodological description in the Methods text. In figure 3e, TdTomato+ cells are shown in the middle panel and P73+ cells in the bottom panel to confirm that TdTomato+ cells are indeed P73+, and thus CRs. It also evidences that the P73+ antibody used overlaps completely with tdTomato. Cells counted around the electrode track were counted based on P73 immunoreactivity in order to use the same criteria for control and mutant animals (control animals do not express tdTomato). The following clarification has been added to the "**Electrode track verification and CRs quantification**" methods section:

P73+ cells around the hippocampal fissure comprised within CA1 were counted in sections containing electrode tracks. P73+ cells around the HF were counted on both sides of the HF: in CA1 SLM as well as in the DG ML. The total number of P73+ cells around the CA1 HF was divided by the total length of the CA1 HF to obtain a cell density value (cells/mm) per section.

Fig. S4: Why was SST expression determined by immunolabeling here and not by ISH as done in Figure 4?

Suitable SST antibodies were obtained only recently.

... Although different studies associate both gain or loss of CRs with convulsive disorders, persistence of CRs has also been described in epilepsy-associated disorders in patients, based on histological studies 15-17....

Again ref. 16 is not appropriate for this statement, please cite here other papers.

The reference has been replaced.

Ref. 75 does not deal with Reelin deficiency in TLE.

We thank the reviewer and the reference has been removed.